# Comprehensive Analysis of the Immune Response to SARS-CoV-2 Epitopes: Unveiling Potential Targets for Vaccine Development

**DOI:** 10.3390/biology14010067

**Published:** 2025-01-14

**Authors:** Huixiong Deng, Yanlei Li, Gefei Wang, Rui Li

**Affiliations:** 1Chaoshan Branch of State Key Laboratory for Esophageal Cancer Prevention and Treatment, Shantou University Medical College, Shantou 515041, China; hxdeng@stu.edu.cn (H.D.); 16ylli3@stu.edu.cn (Y.L.); 2Center of Pathogen Biology and Immunology, Shantou University Medical College, Shantou 515041, China; 3Nursing Research Center, Shantou University Medical College, Shantou 515041, China

**Keywords:** SARS-CoV-2, epitope, structure-based network, immune landscape, broad-spectrum

## Abstract

This study performed a comprehensive meta-analysis of SARS-CoV-2 antibody and T cell epitopes. The goal was to identify candidate immunodominant antigen epitopes that are highly conserved and evolutionarily constrained. It found that immune responses were concentrated in certain regions of the virus proteins. Notably, B cell and CD4^+^ T cell responses were positively correlated with high viral variability, while CD8^+^ T cell responses showed a negative correlation. This study identified highly conserved and evolutionarily constrained SARS-CoV-2 epitopes, which could be crucial for broad-spectrum vaccine development. These findings provide important insights for future research and clinical applications.

## 1. Introduction

The ongoing COVID-19 pandemic, instigated by the SARS-CoV-2, remains a substantial threat to global public health [1,2]. Despite advancements in vaccine development and therapeutic strategies, current SARS-CoV-2 vaccines still face numerous challenges, including viral mutability, immune durability, global accessibility, immune evasion, and safety and side effects. There is a need for the development of broadly protective vaccines that can address the current and future COVID-19 pandemics. Achieving these goals necessitates a deeper understanding of the virus’s immunological properties to inform future research and vaccine strategies [3,4,5,6,7]. The identification and characterization of antigenic epitopes, which are specific segments of the virus recognized by the immune system, are fundamental to understanding the complexities of host–pathogen interactions and enhancing vaccine effectiveness [8,9]. Such knowledge can further facilitate the development of SARS-CoV-2 vaccines, elucidate SARS-CoV-2-associated immunopathology, and serve as a benchmark in the evaluation of various vaccine candidates. Understanding the antigenic epitope landscape of SARS-CoV-2 and identifying highly conserved and evolutionarily constrained immunodominant antigenic epitopes are of great significance in the development of broad-spectrum vaccines. Epitopes of variable pathogens, which are mutationally constrained, are attractive targets for vaccine design, although sequence conservation alone does not reliably identify them. Structure-based network analysis has been proven to be a viable strategy. This method applies network theory to protein structure data, quantifying the topological significance of individual amino acid residues, which has been successfully used in numerous studies for the precise identification of evolutionarily constrained sites and antigenic epitopes [10,11]. This approach allowed us to define mutationally constrained antibody and T-cell antigenic epitopes within the SARS-CoV-2 proteome. These epitopes have the potential to elicit substantial protective immune responses.

The SARS-CoV-2 scientific reports are replete with immunological data, a significant portion of which encompasses studies delineating T-cell and antibody epitopes. The Immune Epitope Database and Analysis Resource (IEDB) offers the scientific community a freely accessible repository of immune epitope data (www.immuneepitope.org accessed on 19 September 2024.) This database includes data extracted from published scientific reports, involving antibody and T-cell data from humans, non-human primates, and rodents, along with other diverse animal species, and encapsulates epitopes associated with all infectious diseases, autoimmunity, transplantation, and allergies. Consequently, the IEDB represents a unique resource for compiling and analyzing immunological data specific to certain pathogens or diseases [12]. N. Magazine et al. analyzed SARS-CoV-2 epitopes in the IEDB database, highlighting the impact of Spike protein mutations on immune evasion. They emphasized that predicting mutations and selecting conserved epitopes are crucial for developing broad-spectrum neutralizing therapies or universal vaccines [13]. In this study, we retrieved SARS-CoV-2 epitope datasets from the IEDB and perform a comprehensive meta-analysis, with the aim of illustrating its immunological landscape and identifying highly conserved and evolutionarily constrained immunodominant epitopes. By integrating and analyzing data from various SARS-CoV-2 isolates, we aim to uncover the structural and sequential relationships of these epitopes and their impact on immune recognition. Our study provides an updated overview of immunological data related to COVID-19, highlighting key patterns and identifying specific regions that require further experimental investigation. By compiling and visualizing SARS-CoV-2 immune reactivity from nearly 690 distinct studies, we render a comprehensive depiction of the scientific community’s endeavors. Our analysis unveils overlapping domains and homologous sequences in SARS-CoV-2 epitopes, thereby spotlighting regions of intense immune activity. These regions are pivotal for antibody binding and the responses of CD4^+^ and CD8^+^ T cells, the latter playing a crucial role in preventing viral infection and virus clearance. Interestingly, we discerned a positive correlation between the epitopes recognized by B cells and CD4^+^ T cells and regions of high viral mutation, suggesting these areas are integral in eliciting a robust immune response. Conversely, CD8^+^ T cell epitopes exhibited a negative correlation with high variability, indicating a preference for more conserved viral regions. From a structural perspective, our network analysis demonstrated a significant positive correlation between CD8^+^ T cell reactivity and residue network connectivity, a relationship not observed in B cell and CD4^+^ T cell reactivity. This discovery underscores the significance of structural constraints in shaping the SARS-CoV-2 immunological landscape. Through the integration of sequence entropy and structural network analysis, we identified highly conserved and evolutionarily restricted epitopes, potentially serving as promising candidates for pan-spectrum vaccine development. Furthermore, employing immunoinformatics, we evaluated the conservation and population coverage of these epitopes across coronaviruses, yielding invaluable insights for vaccine design. Our research unveils key immune reactions associated with viral clearance and highlights potential targets for future vaccine development. These findings establish a solid foundation for propelling SARS-CoV-2 research and optimizing clinical applications, ultimately bolstering global efforts to combat COVID-19 and its variants.

## 2. Materials and Methods

### 2.1. Data Retrieval

For epitopes data, the comprehensive SARS-CoV-2 epitope dataset was accessed on 19 September 2024 by querying the Immune Epitope Database (IEDB, http://iedb.org accessed on 19 September 2024), specifying “SARS-CoV-2” as the source organism, and excluding MHC binding and MHC ligand elution data. This search yielded all available data on antibody (Conformational epitopes and linear epitopes) and T-cell epitope reactivity related to SARS-CoV-2 (including all genotypes, subtypes, and isolates) in both human and non-human (animal model) hosts. Additional searches were conducted using the “T Cell Search” and “B Cell Search” functions on the IEDB website, applying the above extra criteria (such as response phenotype, host organism, SARS-CoV-2 genotype, or detection type) to select subsets of SARS-CoV-2 data. The results of each query were exported as Excel files and further analyzed in this format to generate specific tables and figures. For virus sequence data, a total of 33 representative alpha- and beta-coronavirus genome and protein sequences were retrieved from the NCBI database, encompassing the sub-genus: Setracovirus, Duvinacovirus, Merbecovirus, Sarbecovirus, Hibecovirus, Nobecovirus, Embecovirus, as well as species originating from humans, bats, camels, civet cats, pangolins, and mice. The accession numbers for these sequences are listed in Appendix A. Additionally, a collection of 106 protein sequences representing different lineages of SARS-CoV-2 variants identified by the WHO as Variants of Concern (VOCs) and Variants of Interest (VOIs) were downloaded from the NCBI virus database (Appendix A). Furthermore, SARS-CoV-2 MSA (Multiple Sequence Alignment) sequence data were obtained from the NCBI virus database, with 2000 sequences randomly collected for each protein of the SARS-CoV-2 proteome. These were used for Shannon entropy analysis of the sequences. The MAFFT v7.313 software was employed to align protein and genome sequences derived from humans, bats, pangolins, civets, and camels, with the objective of identifying common factors that induce immunogenicity [14]. For virus structure data, X-ray crystallography structures of Surface spike glycoprotein (PDB ID: 6vsb, 3.46 Å), ORF1ab_nsp3 (PDB ID: 6w9c, 2.70 Å), ORF3a (PDB ID: 6xdc, 2.90 Å), ORF8a (PDB ID: 7jx6, 1.61 Å), Envelope protein (PDB ID: 7tv0, 2.60 Å), Membrane glycoprotein (PDB ID: 7vgr, 2.70 Å), Nucleocapsid (PDB ID: 8fd5, 4.57 Å), and ORF1ab_RdRp, Helicase (PDB ID: 6xez, 3.50 Å) from SARS-CoV-2 were collected from the RCSB PDB database.

### 2.2. Correlation Analysis of Response Frequency (RF) and Entropy in Immune Response

To identify the most researched and frequently recognized epitopes, we used the Immunobrowser tool to calculate the response frequency scores (RF score) for the epitopes. This index reflects the overall recognition frequency of specific epitopes and their specific residues. For a given epitope, the response frequency is calculated based on the number of individuals tested and the number of individuals tested positive, where R = r/t, with r being the total number of responsive donors, and t being the total number of tested donors [15]. Additionally, the lower and upper bounds of the 95% confidence interval (CI) for the response frequency (RF) at each target protein position was calculated. This helps estimate the reliability of the response frequency, determine whether the response frequency is statistically significant, understand and interpret variability and reliability in biological data, and guide further research directions. We calculated the RF score for specific T or B cell epitopes, including those that are broad-ranging (i.e., any host) or specific to certain contexts (for example, T cell tests in humans). The density of SARS-CoV-2 (taxonomy ID No. 2697049) sequence epitopes was visualized using the IEDB Immunobrowser tool [16]. To identify continuous dominant regions, the RF score for each residue was recalculated to represent a sliding window of 10 residues. Concurrently, to measure sequence variability, the Shannon entropy at each position in the multi-protein of SARS-CoV-2 was calculated. Entropy provides a quantification method that can be used to assess the conservation of various positions in a sequence. By calculating the probability of each symbol appearing in a sequence and applying the Shannon entropy formula, the entropy value of the sequence is derived. A low Shannon entropy value indicates that the amino acid at that position is highly conserved across multiple sequences, while a high entropy value indicates a high degree of variation at that position. Compared with sequence conservation analysis based on quantity, sequence conservation analysis based on sequence Shannon entropy considers all possible residue at a location and their probability distribution, thus reflecting the complexity of variation in the sequence, not just the simple mutation frequency. Compared with analysis based on sequence mutation frequency, it has advantages and is widely used in the analysis of biological sequence conservation and variation. Sequence Shannon entropy was determined using the Shannon entropy calculation tool from the Los Alamos National Laboratory HCV database (https://hcv.lanl.gov/content/sequence/ENTROPY/entropy.html accessed on 21 September 2024.)

### 2.3. Correlation Analysis of Response Frequency (RF) and Degree of Structural Network

To analyze functionally important residues in the SARS-CoV-2 proteome, we applied a structural network analysis approach. This method utilizes high-quality protein structural data and network theory to identify topologically significant residues. Residues are evaluated based on their direct and indirect local connectivity through non-covalent interactions, their involvement in interaction networks, and their proximity to known protein ligands. This approach has been widely used in studies to identify key functional residues. Residues with high network connectivity are generally considered to exhibit structural centrality, functional importance, and evolutionary constraints [10]. The following PDB files were used: NSP1 (PDB: 7k7b); NSP2 (PDB: 7MSW); NSP3 ADP-ribose phosphatase domain (PDB: 6W02); NSP3 papain-like protease (PDB: 6W9C); NSP4 (PDB: 7dvp); NSP5 3CL protease (PDB: 6YB7); NSP6 (PDB: 7dvx); NSP7 (PDB: 6M7I, chain C); NSP8 (PDB: 6M7I, chains B, D); NSP9 (PDB: 6W4B); NSP10 (PDB: 6W4H, chain B); NSP12 RNA-dependent RNA polymerase (PDB: 6M7I, chain A); NSP13 (PDB: 6xez); NSP14 (PDB: 7r2v); NSP15 (PDB: 8ud2); NSP16 (PDB: 8tyj); Spike glyoprotein closed conformation (PDB: 6VXX); nucleocapsid RNA binding domain (PDB: 6VYO); nucleocapsid dimerization domain (PDB: 6WJI); ORF3a (PDB: 6XDC); Envelope (PDB: 7tv0); Membrane (PDB: 7vgr); ORF7a (PDB: 6W37); ORF8 (PDB: 7jx6); ORF10 (PDB: 7yc2); Spike glyoprotein open conformation (PDB: 6VYB, 6vsb). Network scores for each protein of the SARS-CoV-2 proteome were calculated using the Ring 4.0 online server [17]. For multimeric proteins, the average degree-based network value at the highest oligomeric state was used.

### 2.4. Population Coverage Analysis

Population coverage calculation was performed using the Population Coverage software (version 3.0) hosted on the IEDB platform to evaluate the distribution of screened CD8^+^ and CD4^+^ T cell epitopes in the global population, considering both HLA-I and HLA-II alleles [18].

### 2.5. Statistical Analyses

All statistical analyses were performed using GraphPad Prism 8.0 software. The primary method of assessing the linear relationship between two continuous variables was the Pearson correlation coefficient. For data not normally distributed, the Spearman rank correlation coefficient was utilized for verification. Comparisons among multiple groups were performed using a one-way ANOVA followed by LSD’s multiple comparison post hoc test. Comparisons between two groups were assessed using unpaired Student’s *t*-tests. For non-normally distributed data, a nonparametric Kruskal–Wallis rank ANOVA with a post hoc Dunn’s test was used. A significance level of *p* < 0.05 was considered statistically significant. The notations * *p* < 0.05, ** *p* < 0.01, *** *p* < 0.001, and **** *p* < 0.0001 were used to indicate the levels of significance. “ns” denotes non-significant results.

## 3. Results

### 3.1. Inventory of SARS-CoV-2 Immune Epitope Data in IEDB

The Immune Epitope Database (IEDB) houses extensive data on positive SARS-CoV-2 antigenic epitopes, offering a comprehensive and intricate portrait of SARS-CoV-2 immune epitopes to date. To identify shared immunodominant SARS-CoV-2 epitopes, we carried out a retrospective meta-analysis of all SARS-CoV-2 epitopes cataloged in the IEDB. As of 19 September 2024, approximately 690 peer-reviewed scientific reports have contributed immunological response data related to SARS-CoV-2 T cell and antibody epitopes. Over 4270 distinct T cell responses and 10,475 different antibody epitopes were scrutinized in more than 63,519 assays (Table 1). The majority of SARS-CoV-2 epitopes were identified in human infections (91.87%), with a smaller proportion derived from mice (3.60%) and non-human primates (3.35%) (Figure 1). Furthermore, the data unveiled different effector response types (antibody, CD4, and CD8) across various SARS-CoV-2 serotypes, including Alpha, Beta, Delta, Epsilon, Gamma, Iota, Kappa, Lambda, Mu, Omicron, and others (Table 2). This collation of epitopes underscores specific immune response trends associated with host and SARS-CoV-2 genotypes, providing a comprehensive breakdown of all SARS-CoV-2 epitopes reported by genotype in the IEDB.

### 3.2. Constructing the Immune Response Map of SARS-CoV-2 Based on Experimental Data

Given that the entire protein of SARS-CoV-2 is composed of approximately 9915 amino acids, there is bound to be significant redundancy and overlap among the 14,745 “epitopes” reported in the peer-reviewed scientific reports. On the one hand, the reported epitopes may be closely related to each other due to sequence variations associated with different SARS-CoV-2 strains and serotypes. Different laboratories and experiments using different but largely overlapping epitopes can also lead to additional overlap. Moreover, there is almost no consistency in protein naming and residue position numbering among different reports. To identify immunoreactive regions within the SARS-CoV-2 proteome, we utilized the IEDB Immuno-browser tool. This tool maps a large number of overlapping and redundant epitopes from different SARS-CoV-2 isolates back to the SARS-CoV-2 reference sequence. By integrating all recorded data on the reference sequence, we generated an RF score. This score accounts for both the positivity rate (the frequency at which a particular residue appears in positive epitopes) and the number of records (the number of independent detections reported) (Appendix A). RF scores serve as an indicator of immune reactivity. Next, we analyzed the epitope maps of most individual proteins of SARS-CoV-2. Overall, the RF score for SARS-CoV-2 antibody epitopes is lower than that for T cell epitopes. The average RF values for all antibody and T cell responses are slightly below 0.05 and 0.25 (corresponding to 5% and 25% percentile values). The 95% confidence intervals (CIs) for antibody response frequencies at each target protein position of SARS-CoV-2 demonstrate high precision, indicating strong reliability in the response frequencies (Appendix A). For CD4^+^ T cell response frequencies, the 95% CIs show high precision within structural and accessory protein regions, suggesting high confidence in these frequencies (Appendix A). The CD8^+^ T cell response frequencies also exhibit high-confidence intervals distributed across each target protein position of SARS-CoV-2 (Appendix A). Figure 2 shows the antibody and T cell RF scores for individual SARS-CoV-2 proteins. The overall antibody response RF score for ORF1ab is low. CD4^+^ T cells show significant responses in the nsp1, nsp3, nsp7, nsp8, nsp13 proteins. CD8^+^ T cells, except for nsp12 and nsp15, which have a lower response frequency, show significant responses distributed along their entire length, with multiple significant immune response clusters (Figure 2A). Further analyses details are in the Appendix B data (Figure A1). The spike glycoprotein plays a crucial role in antibody immunity and is the core target for vaccine development and antibody therapy. It elicits a strong immune response, with the largest antibody response clusters mainly concentrated in the RBD (amino acids 319–541, RF scores 0.09–0.36), FP (amino acids 788–806, RF scores 0.17–0.34), HR2 (amino acids 1163–1213, RF scores 0.06–0.28), and CP (amino acids 1237–1273, RF scores 0.05–0.32) structural domains. The sharpest peaks of antibody response are at aa820, followed by aa846, aa567, aa1158, aa1268. The immune response domains of neutralizing antibodies are mainly in the RBD structural domain aa334–552 (RF scores 0.09–0.36) (Appendix A). In contrast, higher CD4^+^ and CD8^+^ T cell responses are distributed along the protein length. Some CD4^+^ and CD8^+^ T cell responses are observed in SP (aa1–14), sharing a cluster, and additional CD4^+^ T cell responses are also observed at aa236–249 (RF scores 0.45–0.48). For the Spike glycoprotein, CD4^+^ and CD8^+^ T cell responses share a cluster between aa300 and 600 (also present in antibody responses), and there is another significant CD4^+^ T cell response cluster between aa850 and 1050 (Figure 2B). Two robust antibody response clusters were observed in the ORF3a protein structure, specifically at the N-terminal and C-terminal regions, spanning aa13–30 (RF scores 0.10–0.18) and aa251–275 (RF scores 0.10–0.21). The frequency of ORF3a CD4^+^ T cell responses was comparatively low, with a single intense response cluster only present at the C-terminal region aa261–275 (RF scores 0.12–1.00). However, significant CD8^+^ T cell responses were evident throughout the entire ORF3a length, with a response cluster at the C-terminal of ORF3a shared with CD4^+^ T cell and antibodies (Figure 2C). For the Envelope protein, antibody response clusters were concentrated at aa34–75 (RF scores 0.10–0.17). CD4^+^ T cell had two major response clusters at aa25–40 and aa55–70, while CD8^+^ T cell showed noticeable responses in the region of aa12–45, sharing a cluster with CD4^+^ T cell (Figure 2D). The Membrane protein exhibited antibody response regions at both the N-terminal and C-terminal, specifically at aa1–22 (RF scores 0.10–0.51) and aa154–222 (RF scores 0.09–0.22). CD4^+^ T cell responses were lower in the N-terminal region, with two response peaks in the latter half of the protein at aa146–158 and aa173–190. In contrast, CD8^+^ T cell epitopes had two response peaks in the N-terminal region at aa29–30 and aa83–85 (Figure 2E). For ORF6, antibody responses were relatively low, with an RF score < 0.08. Both CD4^+^ and CD8^+^ T cell responses were observed in the N-terminal region, sharing a response cluster (Figure 2F).

The antibody and T cell responses presents a stark contrast for ORF7a and ORF7b. Antibody responses were negligible, whereas intense CD4/CD8 T cell responses were observed at both the N-terminal and C-terminal regions of ORF7a and ORF7b (Figure 3A,B). The antibody and T cell responses for ORF8 is notable across the entire protein, with CD4^+^ T cell response distributed throughout the protein and two response peaks for CD8^+^ T cell at the N-terminal (Figure 3C). For the Nucleoprotein, both antibody and T cell responses are distributed throughout the protein, demonstrating significant reactivity (Figure 3D). The antibody and T cell responses for the region aa1–42 of ORF9 is pronounced, sharing a common response cluster (Figure 3E). Lastly, for the ORF10 protein, the antibody and T cell responses presents a stark contrast. The frequency of antibody responses across the entire protein is low, while a strong CD8^+^ T cell response cluster is present at the N-terminal of the ORF10 protein, with some reactivity from CD4^+^ T cell in the same region (Figure 3F). These data suggest that there was some correlation between the regions of SARS-CoV-2 proteins recognized by antibody, CD4^+^, and CD8^+^ T cell responses. Upon conducting a correlation analysis of the RF scores for the three responses across the entire length of multiple proteins, we found a moderate positive correlation between B-cell responses and CD4^+^ T cell responses (Pearson correlation coefficient R value 0.4, *p*-value < 0.05). However, there was mostly no correlation between B-cell responses and CD8^+^ T cell responses or between CD4^+^ and CD8^+^ T cell response variables (Pearson correlation coefficient R values ranged from −0.15 to 0.08). In summary, the above analysis provides a detailed map of SARS-CoV-2-specific immune responses and indicates that different regions of SARS-CoV-2 are independently targeted by immune responses.

### 3.3. Integration of Sequence Shannon Entropy and Structure-Based Network Analysis Reveals Highly Conserved and Evolutionarily Constrained Immune Response Regions in SARS-CoV-2

Immune pressure can lead to high variability in pathogen sequences, particularly for rapidly mutating RNA viruses. Therefore, accurately identifying immunodominant epitopes with evolutionary constraints is crucial for developing durable vaccines. Studies have shown that structure-based network analysis can predict regions in SARS-CoV-2 that are relatively mutation-resistant or prone, identifying highly conserved residues superior to traditional sequence conservation metrics in detecting mutation-intolerant residues in SARS-CoV-2 T cell epitopes, given the continuously evolving viral sequence landscape [9,11]. In this study, utilizing high-quality structural data of the SARS-CoV-2 proteome, we employed a similar strategy to analyze the network connectivity of residues within SARS-CoV-2 proteins performing different functions. An example of residue network analysis for the Spike glycoprotein trimer is shown in Figure 4A, with analyses for other SARS-CoV-2 proteins presented in Appendix A. Through the structural network analysis, residues in SARS-CoV-2 that are resistant to mutation have been discerned. To evaluate the mutation tolerance of these highly networked residues, residue network scores were categorized into three groups (0–2, 2–4, >4) and juxtaposed with the sequence entropy values of SARS-CoV-2. Our results align with the findings of Anusha Nathan et al. [11], demonstrating a negative correlation between the topological importance of network metrics and mutation frequency, as quantified by Shannon entropy (Figure 4B). This implies that structural network analysis can serve as a valuable tool for assessing the mutation tolerance of highly networked residues and for identifying regions of topological significance. Subsequently, we established a correlation between the RF scores for antibodies and T cells and the Shannon entropy based on sequence. On a sequence level, there exists a positive correlation between epitopes recognized by B cells and CD4^+^ T cells and high sequence variability. This suggests that regions with a high degree of variability are typically associated with elevated RF scores, as evidenced by a Pearson correlation coefficient (R) of 0.24 and 0.13, respectively, and a *p*-value < 0.05 (Figure 4C,D). Conversely, epitopes recognized by CD8^+^ T cells demonstrate a negative correlation with high variability, as evidenced by a Pearson correlation coefficient (R) of −0.06 and a *p*-value < 0.05 (Figure 4E). This suggests that conserved regions are more likely to be universally recognized. Upon correlating the centrality of the residue network with the RF scores for antibodies and T cells, we discovered no significant correlation between network centrality and the reactivity of epitopes for either B cells or CD4^+^ T cells, with *p*-values of 0.197 and 0.216, respectively (Figure 3F,G). Contrarily, the reactivity of CD8^+^ T cells exhibited a significant positive correlation with the centrality of the residue network, as evidenced by a Pearson correlation coefficient (R) of 0.08 and a *p*-value < 0.05 (Figure 4H). This suggests that epitopes of CD8^+^ T cells with high RF scores are likely to be located in regions with highly networked residues. Consistent with the sequence entropy analysis, it appears infeasible to identify candidate epitopes for B cells and CD4^+^ T cells that are both highly conserved, evolutionarily constrained, and highly immunoreactive. To screen for highly conserved and evolutionarily constrained B cell and CD4^+^ T cell epitopes, a balance among these critical parameters must be struck. However, for CD8^+^ T cell epitopes, it is feasible to find candidates that meet these criteria. By considering residues with RF scores above 0.3, we defined immunodominant regions. B cell epitopes are primarily associated with SARS-CoV-2 structural and accessory proteins (Figure 5A). CD4^+^ T cell epitopes are mainly located in nsp1, nsp3, nsp5, nsp13, nsp14, and structural/accessory proteins (Figure 5B). CD8^+^ T cell epitopes are broadly distributed across the entire SARS-CoV-2 proteome (Figure 5C). Aligning SARS-CoV-2 residue network scores with sequence entropy values revealed numerous linear regions where highly networked and conserved B cell and T cell epitopes could be identified (Figure 5D,E).

### 3.4. The Potential Immunological Targets Exhibit High Coverage Across Multiple Variants of Concern (VOCs) and Variants of Interest (VOIs) of SARS-CoV-2, as Well as Other Coronaviruses

We selected twelve immunodominant B-cell epitopes from the epitope dataset by integrating RF scores, sequence Shannon entropy, and structural network indices (Appendix A). Through sequence alignment and cluster analysis, we found that epitope B3 (spike glycoprotein, aa862–876) is completely conserved across all 106 variants of VOCs and VOIs. The remaining 11B-cell epitopes exhibit high conservation among these variants, with most strains being 100% conserved and a few strains having at least 75% conservation (Appendix A). Notably, we found that ten B-cell epitopes are 100% conserved in two or more H-CoVs and SL-CoVs derived from pangolins and bats, including GX-P4L, GX-P5E, GX-P5L, P2V, P1E, MP789, Bt-CoV recombinants, Rs672, W1V1, and YNLF_31C. Furthermore, ten B-cell epitopes have at least 70% conservation in SARS-CoV and SL-CoVs (such as SARS-CoV-Urbani, HCoV-OC43), as well as SL-CoVs from bats, pangolins, and camels (such as Civet007, BtCoV-HKU4, BtCoV recombinants, Rs672, WIV1, Bat Hp-BetaCoV, Pipistrellus bat coronavirus HKU5 isolate YD13403, BtCoV-WIV16, BtCoV-RATG13, Rousettus BtCoV-GCCDC1, YNLF_31C, MERS-CoV, HCoV-EMC [MERS-related coronavirus strain]; GX-P1E, GX-P5E, GX-P4L, GX-P1E, GX-P5L, GX-P5E, GX-P2V, MP789) (Appendix A). For CD4^+^ T-cell epitopes, we selected 16 immunodominant CD4^+^ T-cell epitopes from the epitope dataset (Appendix A). These epitopes demonstrated strong binding abilities with various MHC-II alleles and induced the release of specific cytokines, such as IFN-γ, TNF-α, IL-2, IL-5, and IL-4. Among these candidate epitopes, seven are located in the spike glycoprotein, and six in the nucleocapsid protein. Conservation analysis revealed that Th1, Th5, Th9, Th11, Th15, and Th16 epitopes, corresponding, respectively, to the membrane glycoprotein (aa176–190), nucleocapsid protein (aa311–325), spike glycoprotein (aa866–880), ORF8a (aa43–57), spike glycoprotein (aa896–910), and nucleocapsid protein (aa107–121), are 100% conserved across all 106 variants of SARS-CoV-2 (Appendix A). Among the Th-cell epitopes, sixteen are 100% conserved in two or more H-CoVs and SL-CoVs. However, the SL-CoV strains from civets and the MERS-CoV strains isolated from camels showed lower conservation (Appendix A). For CD8^+^ T-cell epitopes, we identified 18 CD8^+^ T-cell epitopes (Appendix A), which originate from the spike glycoprotein, nucleocapsid phosphoprotein, and open reading frames (ORF3a and ORF1ab) of SARS-CoV-2. The highest similarity was observed across 106 strains of SARS-CoV-2, eight strains of previous HCoVs, and 25 strains of animal SL-CoV isolated from bats, civets, pangolins, rodents, and camels. CTL1, CTL2, CTL3, CTL4, CTL5, CTL7, CTL9, CTL14, CTL15, and CTL16 exhibited 100% conservation across all 106 variants of concern (VOCs) and variants of interest (VOIs) of SARS-CoV-2 (Appendix A). All eighteen CTL epitopes were 100% conserved in two or more H-CoVs and SL-CoVs, including the four ‘common cold’ coronaviruses that caused previous epidemics (HCoV-OC43, HCoV-229E, HCoV-HKU1, and HCoV-NL63), as well as SL-CoVs isolated from bats and pangolins (Appendix A). The regions of the immunodominant candidate epitopes for CD8^+^ T cells were highly conserved and had high network indices, suggesting evolutionary constraints. The alignment logo plots of all candidate antibody and T-cell reactive epitopes and their locations in different proteins are shown in Figure 6A. Moreover, our analysis showed that all conserved epitopes could either cross-react among different strains or are unique to different zoonotic coronavirus strains (Figure 6B–D). Lastly, our computational model-based predictions indicated that these candidate epitopes exhibited non-toxic effects on the host (Appendix A).

### 3.5. The MHC Polymorphism Restricted of Candidate T-Cell Epitopes, Providing Global Population Coverage

A given T-cell epitope can only elicit a response in individuals who express the corresponding specific MHC molecule. Therefore, selecting multiple epitopes with different HLA binding specificities can significantly enhance coverage of the patient population targeted by epitope-based vaccines or diagnostics. We have verified that all T-cell epitopes are Pan-DR helper T cell epitopes and Pan-HLA-A epitopes. A total of CTL candidate epitopes that can be presented to CD8^+^ T cells by 32 of the most prevalent HLA-A alleles. Additionally, we have identified Th candidate epitopes restricted by 50 of the most prevalent HLA-DRB/DQB/DQA/DPB alleles, as shown in Appendix A. It is worth noting that these epitopes can cross-bind with high or intermediate affinity to the aforementioned HLA-A alleles. Notably, the HLA-A2-restricted candidate epitopes can also be cross-presented by mouse H-2K/Db molecules, which provide feasibility for mouse-based animal model to validate vaccine immunogenicity. Population coverage analysis reveals that the selected T-cell epitopes provide coverage for 100% of the global population, as well as individual regions such as Northeast Asia, Europe, North America, Southeast Asia, East Africa, and China. East Asia, representing the main epidemic zone for SARS-CoV-2, has a coverage rate of 99% (Table 3). The allelic frequency data confirm the global distribution characteristics of the selected T-cell epitopes, making them suitable for designing candidate protein vaccines.

## 4. Discussion

Existing SARS-CoV-2 vaccines are widely used globally, including mRNA vaccines (Pfizer and Moderna) [19,20], viral vector vaccines (AstraZeneca and Johnson & Johnson) [21,22], inactivated vaccines (such as Sinovac and Sinopharm) [21,22], and recombinant protein vaccines (Novavax) [23,24]. These vaccines have achieved significant success in reducing infection rates and severe disease. However, with the constant mutation of the virus, especially the emergence of new variants that can partially evade immune responses, the effectiveness of existing vaccines may be challenged. Developing broad-spectrum vaccines by targeting highly conserved and evolutionarily constrained immunodominant epitopes may be an effective strategy. In this study, we systematically characterized the landscape of antibody and cellular immune responses to SARS-CoV-2 through a meta-analysis, uncovering the immunological features of SARS-CoV-2 and offering a fresh perspective for comprehending its immune response within the host. The latest data, derived from hundreds of distinct published reports encompassing thousands of unique molecular structures, scrutinized over 63,519 disparate experiments, all compiled within the IEDB analysis resources. The epitope data inventory revealed that while the majority of records unsurprisingly originated from humans (91.87%), a substantial number also stemmed from murine (roughly 3.60%) and primate systems (approximately 3.36%). Moreover, the inventory disclosed a significant imbalance in the coverage of different effector response types (antibody, CD4^+^, and CD8^+^ T cell) across various SARS-CoV-2 serotypes, highlighting the necessity to generate additional data in these domains. The epitope data analyzed from various sources introduces variability due to differences in experimental methods, data quality, and biological factors. The Immune Epitope Database (IEDB) mitigates these challenges through data standardization, quality control [12], and tools like the ImmunomeBrowser, which visualizes immune responses and knowledge gaps [16]. RF score prioritizes candidate epitopes based on response frequency, aiding vaccine development, but is limited by data biases, population diversity, and static datasets. To address this, 95% confidence intervals for RF were calculated to assess reliability and guide further research. Future efforts should focus on refining data categorization to extract deeper insights.

We determined the antigens of SARS-CoV-2 that prevail in different response types and charted the comprehensive response patterns of various antigens. Our investigation revealed that specific regions within SARS-CoV-2 proteins display heightened activity in the immune responses of antibodies, CD4^+^ T cells, and CD8^+^ T cells. Identifying these regions is vital for the development of more effective vaccines. Notably, the epitopes recognized by B cells and CD4^+^ T cells are positively associated with high viral variability, suggesting that these areas could be hotspots for the virus to evade immune surveillance. The virus’s mutations within these sites could represent a strategy to circumvent host immune pressure [25,26,27,28,29]. However, these highly variable areas also serve as potent inducers of immune responses, indicating the need to strike a balance between antigen variability and immunogenicity in vaccine design. Conversely, the epitopes recognized by CD8^+^ T cells are negatively associated with high variability, implying that CD8^+^ T cell responses may favor targeting the virus’s conserved regions. These conserved regions often fulfill a crucial role in the virus’s function and are subject to stronger evolutionary pressure, thus mutating less frequently. This characteristic of CD8^+^ T cells underscores their significant role in antiviral immunity, particularly in tackling variant strains. T cell-mediated adaptive immunity is fundamental in immune surveillance and host control of infectious diseases [8,30]. Many viruses can circumvent cellular immunity by mutating HLA class I restricted epitopes [31,32]. The mutational constraints of variable pathogen epitopes present promising targets for vaccine design, but their identification through sequence conservation was not reliable [10]. Pinpointing mutation-intolerant regions within viral proteins through protein network analysis, and subsequently identifying highly networked epitopes within the viral proteome that are mutation-intolerant, offer viable strategies for crafting effective broad-spectrum vaccines against both prevalent and emerging variant strains [8,25,26,27,28,33]. The identification of virus-specific, non-conservative, major immunogenic T cell epitopes is crucial for the development of efficacious specific diagnostic tools. Through the structural network analysis, we discerned a significant positive correlation between CD8^+^ T cell responses and the degree of residue network connectivity. This discovery suggests that CD8^+^ T cells might favor recognizing those core areas that have crucial functions within the viral structure. Highly connected residues commonly reside in the functional core of proteins, participating in key molecular interactions, and are thus evolutionarily conserved. Recognizing these areas not only aids in understanding the mechanism of action of CD8^+^ T cells, but also offers strategies for vaccine design targeting these conserved areas. Our correlation analysis with sequence entropy and structural network complexity revealed that B cell epitopes and CD4^+^ T cell responses are positively correlated with sequence entropy but not with structural network complexity. Notably, CD8^+^ T cell immune response epitopes showed significant correlations with both sequence entropy and structural network complexity, highlighting the feasibility of targeting highly conserved and evolutionarily constrained CD8^+^ T cell epitopes. By identifying linear regions of high conservation and structural network complexity in the SARS-CoV-2 proteome, we selected CD8^+^ T cell candidate epitopes that are both highly conserved and evolutionarily constrained. Additionally, we selected linear B cell epitopes capable of inducing neutralizing antibodies, as well as highly conserved and immunogenic CD4^+^ T cell epitopes. Some of these epitopes have been demonstrated to have structural constraints that limit SARS-CoV-2 evolution and are potentially invariant CD4^+^ and CD8^+^ epitopes, making them excellent candidates for next-generation vaccines [34,35].

Additionally, we assessed the cross-protection potential of the identified epitopes, integrating immunoinformatics analysis. The finding that many B and T cell epitopes are highly conserved between SARS-CoV-2 and other coronaviruses is crucial. Vaccine strategies tailored to elicit immune responses to these conserved epitope regions could engender immunity that offers not only cross-protection against β-coronaviruses but also displays relative resistance to the ongoing evolution of the virus. Lastly, we evaluated the population coverage of the identified epitopes. By forecasting the immune responses of these epitopes in diverse populations, we can more effectively design vaccines with broad applicability and sustained protective effects. This methodology aids in identifying the most promising candidate epitopes during the early stages of vaccine development, optimizing the vaccine’s immunogenicity and safety. Our study identifies conserved immunodominant antigenic epitopes across multiple zoonotic coronaviruses and SARS-CoV-2 variants, highlighting their potential as effective immune targets. These findings provide a basis for developing vaccines that elicit broad immune responses. This study also emphasizes the role of these epitopes in interacting with the host immune system, particularly in inducing neutralizing antibodies and T-cell responses. By analyzing epitope conservation across different strains, we offer new insights into immune evasion mechanisms, which are critical for designing vaccines capable of addressing viral mutations. These findings not only guide vaccine development but also serve as a theoretical foundation for future experimental validation.

Despite the valuable insights our research offers, there are some limitations. Firstly, the meta-analysis is dependent on existing public data, which might be influenced by data quality and completeness. Future research should persist in exploring the immune response characteristics of these epitopes in diverse populations, particularly considering differences across various age groups, genders, and ethnicities. Moreover, with the advent of new SARS-CoV-2 variants, ongoing monitoring of these epitopes’ conservation and immunogenicity is essential. This will aid in the development of broad-spectrum vaccines capable of tackling future variants, thereby providing long-term protection for global public health. Future work should validate the candidate epitopes experimentally to aid vaccine and diagnostic tool development. This involves examining the cross-recognition of these epitopes across variants using immunological assays. Also, their immunogenicity and protection range should be assessed in animal models. Furthermore, based structural biology studies to understand how these epitopes interact with the immune system. And lastly, ensure no adverse reactions in preclinical toxicology and safety evaluations. In summary, this study offers fresh insights into the immune response mechanisms of SARS-CoV-2 and vaccine design. By identifying and validating conservative and highly immunogenic epitopes, we have established a foundation for confronting the continuous health threats posed by SARS-CoV-2 and its variants.

## 5. Conclusions

This study pinpointed the key antigenic epitopes of SARS-CoV-2 through a meta-analysis and shed light on the role of different immune responses in areas of viral variability and conservation. We discovered that CD8^+^ T cells show a preference for recognizing conservative regions, which hold substantial potential in vaccine design. Structural network analysis further affirmed the central role of these regions in viral functionality. Coupled with immunoinformatics analysis, we assessed the wide applicability of these epitopes, offering practical guidance for the creation of broad-spectrum vaccines. While our findings lay a robust foundation for vaccine development, they still require further experimental validation. In summary, this study imparts key insights into the immune mechanisms of SARS-CoV-2 and the design of broad-spectrum vaccines, contributing to the mitigation of health challenges posed by future variants.

## Figures and Tables

**Figure 1 biology-14-00067-f001:**
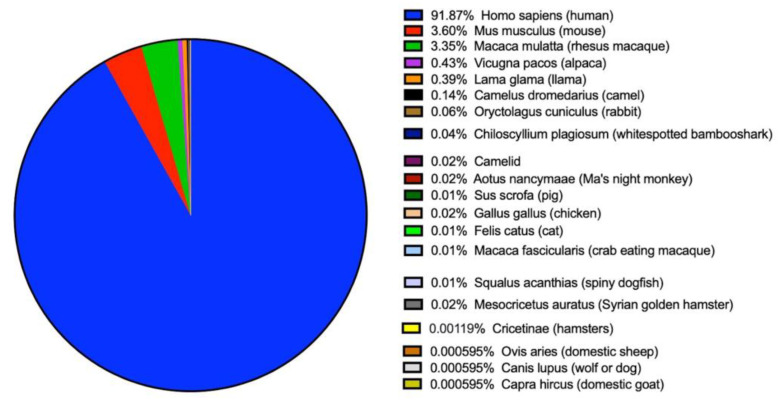
Distribution of epitopes across host species. Data represent the percentage of epitopes identified in each host species to date.

**Figure 2 biology-14-00067-f002:**
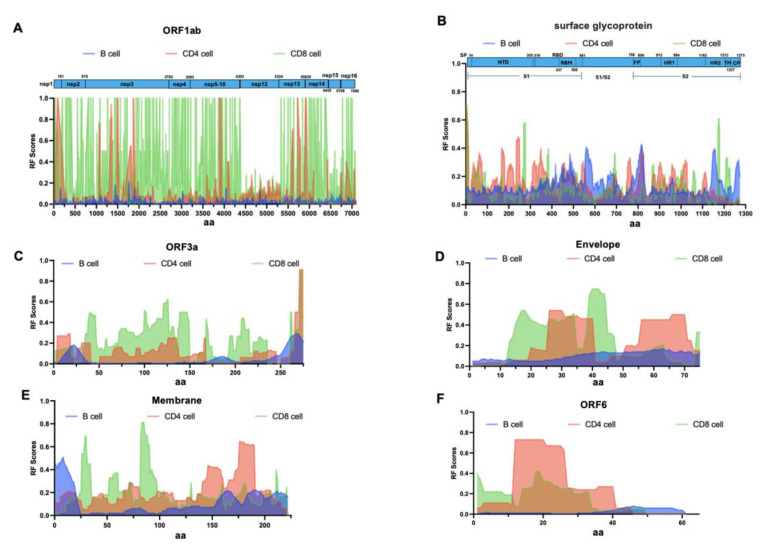
Detailed reactivity maps for SARS-CoV-2 proteins. Data represent individual RF scores for antibody, CD4^+^, and CD8^+^ T cell responses plotted for each antigen translated from the SARS-CoV-2 2019-nCoV reference polyprotein: (**A**) ORF1ab (UniProtP0DTD1), (**B**) Spike glycoprotein (UniProtP0DTC2), (**C**) ORF3a (UniProtP0DTC3), (**D**) Envelope (UniProtP0DTC4), (**E**) Membrane (UniProtP0DTC5), (**F**) ORF6 (UniProtP0DTC6). RF scores (frequency values in the 0–1 range) are shown on the Y-axis, while the amino acid positions of the SARS-CoV-2 proteome are displayed on the X-axis. Blue indicates antibody responses (both conformational and linear epitopes). Red represents CD4^+^ T cell responses, and green indicates CD8^+^ T cell responses. The lower and upper bounds of the 95% confidence interval (CI) for the response frequency (RF) at each target protein position can be found in Appendix A.

**Figure 3 biology-14-00067-f003:**
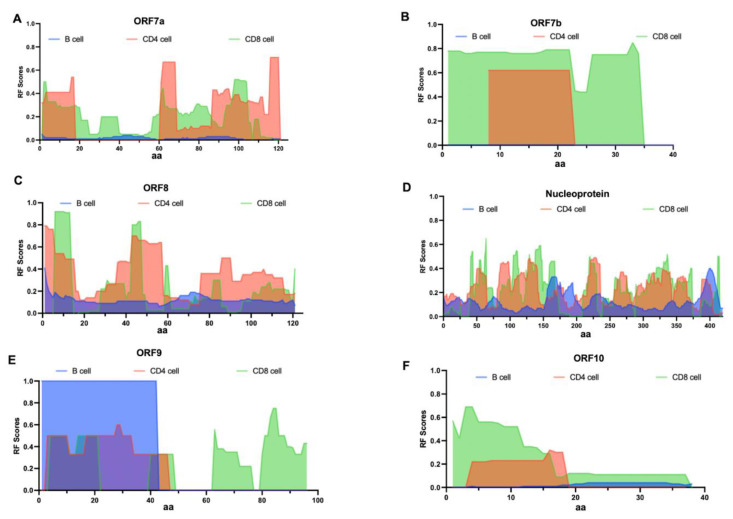
Detailed reactivity maps for SARS-CoV-2 proteins. Data represent individual RF scores for antibody, CD4^+^, and CD8^+^ T cell responses plotted for each antigen translated from the SARS-CoV-2 2019-nCoV reference polyprotein: (**A**) ORF7a (UniProtP0DTC7), (**B**) ORF7b (UniProtP0DTD8), (**C**) ORF8 (UniProtP0DTC8), (**D**) Nucleoprotein (UniProtP0DTC9), (**E**) ORF9 (UniProtP0DTD2), (**F**) ORF10 (UniProtA0A663DJA2). RF scores (frequency values in the 0–1 range) are shown on the Y-axis, while the amino acid positions of the SARS-CoV-2 proteome are displayed on the X-axis. Blue indicates antibody responses (both conformational and linear epitopes). Red represents CD4^+^ T cell responses, and green indicates CD8^+^ T cell responses. The lower and upper bounds of the 95% confidence interval (CI) for the response frequency (RF) at each target protein position can be found in Appendix A.

**Figure 4 biology-14-00067-f004:**
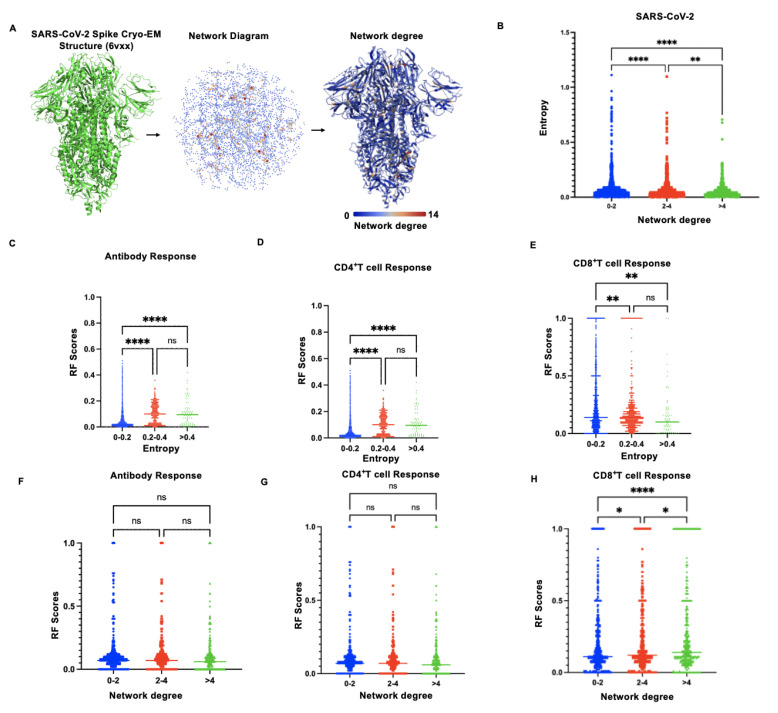
Sequence and structure-based network analysis reveals highly conserved and evolutionarily constrained immunodominant regions in SARS-CoV-2. (**A**) Schematic of the structure-based network analysis for the closed spike glycoprotein trimer (PDB: 6VXX), where amino acid residues are depicted as nodes and non-covalent interactions as edges. The edge width indicates interaction strength, while the color intensity and node size represent relative network scores. (**B**) Comparison of SARS-CoV-2 amino acid network scores (categorized as network score ranges: 0–2, 2–4, and >4) with SARS-CoV-2 sequence entropy. (**C**–**E**) Comparison of SARS-CoV-2 amino acid entropy scores (categorized as entropy score ranges: 0–2, 2–4, and >4) with SARS-CoV-2 B, CD4^+^ and CD8^+^ T cell epitope RF Scores. (**F**–**H**) Comparison of SARS-CoV-2 amino acid network scores (categorized as network score ranges: 0–2, 2–4, and >4) with SARS-CoV-2 B, CD4^+^ and CD8^+^ T cell epitope RF scores. Data comparison between two groups was performed using an unpaired *t*-test. A one-way analysis of variance (ANOVA) followed by Fisher’s LSD multiple comparisons was used. Calculated *p*-values are as follows: * *p* < 0.05; ** *p* < 0.01; **** *p* < 0.0001; ns, not significant.

**Figure 5 biology-14-00067-f005:**
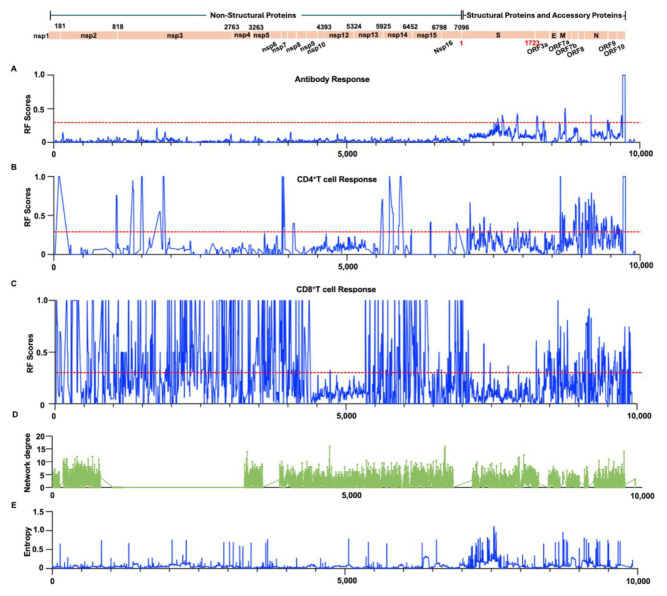
Comparison of SARS-CoV-2 antibody RF scores (**A**), CD4^+^ T cell RF scores (**B**), and CD8^+^ T cell RF scores (**C**) along the length of the SARS-CoV-2 proteins with network scores (**D**) and sequence entropy values (**E**). Regions with RF scores above 0.3 (indicated by red dashed lines) were used as criteria for selecting candidate antigenic epitopes.

**Figure 6 biology-14-00067-f006:**
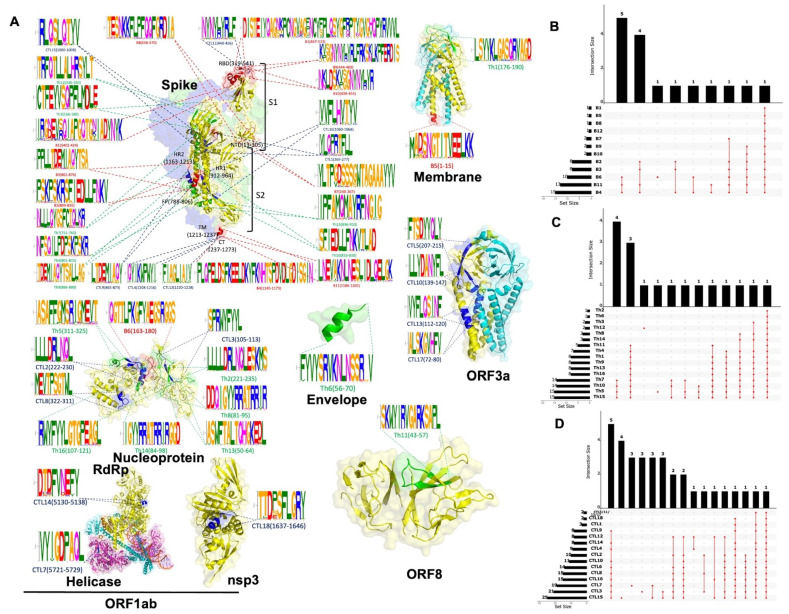
Candidate epitopes mapping to SARS-CoV-2 structural and non-structural proteins and the conserved relationship with different coronavirus strains. (**A**) Mapping of candidate epitopes to structures of spike glycoprotein (PDB ID: 6vsb, 3.46 Å), ORF1ab_nsp3 (PDB ID: 6w9c, 2.70 Å), ORF3a (PDB ID: 6xdc, 2.90 Å), ORF8a (PDB ID: 7jx6, 1.61 Å), Envelope protein (PDB ID: 7tv0, 2.60 Å), Membrane glycoprotein (PDB ID: 7vgr, 2.70 Å), Nucleocapsid (PDB ID: 8fd5, 4.57 Å) and ORF1ab_RdRp, Helicase (6xez, 3.50 Å). Every candidate epitope motif’s logo and the position of structure. The red, blue, and green colors represent B cell, CD4^+^, and CD8^+^ T cell epitopes, respectively. (**B**–**D**) Conservativeness analysis of B, CD4^+^, CD8^+^ T cell candidate epitopes in 33 zoonotic coronavirus strains. The left bar graph indicates the number of strains corresponding to fully conserved or cross-reactivity epitopes for each candidate epitope; the upper bar graph suggests the number of virus strains mapped to the same strain among different epitopes; and the middle-dotted line connecting graph refers to the number of strains corresponding to the intersecting viruses among different epitopes.

**Table 1 biology-14-00067-t001:** Summary of immune epitope data for SARS-CoV-2.

		References	Total Structures	Epitopes	Assay
B and T cell (all)		690	18	18,320	63,519
Tell	Th	82	13	1696	5146
	CTL	154	14	2719	7467
Antibody	Discontinuous	386	6	1253	15,983
	Linear peptide	144	14	9222	18,710

**Table 2 biology-14-00067-t002:** SARS-CoV-2 epitope distribution by genotype.

SARS-CoV-2 Genotype Unspecified	Epitopes	Total Positive	Antigen	Assay	Receptor	Reference
SARS-CoV-2 Alpha	Totall T	32	4	109	0	6
	Th	19	1	35	0	2
	CTL	31	4	73	0	5
	Total B	16	1	111	5	7
	Discontinuous	10	1	101	5	6
	Linear	6	1	10	0	1
SARS-CoV-2 Beta	Totall T	19	2	70	0	2
	Th	18	1	36	0	1
	CTL	19	2	34	0	2
	Total B	85	1	615	47	22
	Discontinuous	85	1	615	47	22
	Linear	0	0	0	0	0
SARS-CoV2 Epsilon	Totall T	0	0	0	0	0
	Total B	2	2	29	2	1
	Discontinuous	2	2	29	2	1
	Linear	0	0	0	0	0
SARS-CoV2 Gamma	Totall T	28	2	84	0	3
	Th	22	1	39	0	1
	CTL	28	2	45	0	3
	Total B	2	1	10	2	1
	Discontinuous	2	1	10	2	1
	Linear	0	0	0	0	0
SARS-CoV2 Iota	Totall T	5	4	5	0	1
	Th	0	0	0	0	0
	CTL	5	4	5	0	1
	Total B	0	0	0	0	0
SARS-CoV2 Kappa	Totall T	0	0	0	0	0
	Total B	3	1	15	3	1
	Discontinuous	3	1	15	3	1
	Linear	0	0	0	0	0
SARS-CoV2 Lambda	Totall T	1	1	1	0	1
	Th	1	1	1	0	1
	CTL	0	0	0	0	0
	Total B	3	1	12	0	1
	Discontinuous	3	1	12	0	1
	Linear	0	0	0	0	0
SARS-CoV2 Mu	Totall T	0	0	0	0	0
	Total B	1	1	20	1	1
	Discontinuous	1	1	20	1	1
	Linear	0	0	0	0	0
SARS-CoV2 Omicron	Totall T	255	7	455	0	18
	Th	148	3	210	0	8
	CTL	144	7	178	0	12
	Total B	156	2	3525	67	75
	Discontinuous	149	2	3463	65	70
	Linear	7	1	62	2	5
SARS-CoV2 Other	Totall T	370	11	960	1	23
	Th	140	10	414	0	10
	CTL	203	9	511	1	19
	Total B	1566	10	5880	119	83
	Discontinuous	189	2	2463	95	70
	Linear	1377	9	3417	24	15

**Table 3 biology-14-00067-t003:** Population coverage calculation of the helper T lymphocyte (HTL) epitope and cytotoxic T-lymphocyte (CTL) epitope specific allele.

Population/Area	MHC Class	Coverage ^a^	Average_Hit ^b^	PC90 ^c^
World	Combined	100.00%	25.45	16.36
East Asia	Combined	99.99%	21.12	12.48
Northeast Asia	Combined	100.00%	19	10.63
Europe	Combined	100.00%	28.93	19.95
North America	Combined	100.00%	27.54	18.6
Southeast Asia	Combined	100.00%	18.92	10.58
East Africa	Combined	100.00%	20.89	12.76
China	Combined	100.00%	18.98	10.6

^a^ Projected population coverage. ^b^ Average number of epitope hits/HLA combinations recognized by the population. ^c^ Minimum number of epitope hits/HLA combinations recognized by 90% of the population.

## Data Availability

All data generated or analyzed during this study are included in this published article [and its Appendix A].

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
