# Peer review of "Comprehensive Analysis of the Immune Response to SARS-CoV-2 Epitopes: Unveiling Potential Targets for Vaccine Development"

_biology, 2025, doi:10.3390/biology14010067_

Round 1
Reviewer 1 Report
Comments and Suggestions for Authors
The Manuscript titled “Comprehensive Analysis of the Immune Response to SARS-CoV-2 Epitopes: Unveiling Potential Targets for Vaccine Development” by Deng et al. presents a robust meta-analysis of SARS-CoV-2 epitopes, integrating sequence and structural analyses to identify potential vaccine targets. It is timely and addresses an important topic. However, some concerns must be addressed to increase clarity, scientific consistency, and accessibility.
1. The manuscript uses sentences that are too complex and includes technical terms like "sequence Shannon entropy" and "structural network correlation," which make it hard to read. "Response frequency (RF) scores" in the method section. Please simplify the language and explain specialized terms early in the text to make the study more straightforward for a wider immunology audience.
2. Add a brief summary of the current challenges in SARS-CoV-2 vaccine research to highlight how this study contributes to addressing them.
3. The manuscript does not discuss the vaccines currently being used for SARS-CoV-2. It would help explain how this study improves existing vaccine strategies and emphasize its importance in tackling emerging variants.
4. The methods section is detailed but contains many technical terms that can take time to follow. Simplifying or providing extra explanations for terms like "response frequency (RF) scores" would make it easier to understand.
5. The results cover a lot of information but are presented in an overwhelming way. Breaking them into sections like "Epitope Analysis," "Population Coverage," and "Structural Insights" could improve clarity and readability.
6. The figures, such as the reactivity maps and network analyses, are complex and challenging to interpret. Consider splitting them into smaller, focused panels and adding clearer explanations in the figure legends.
7. Include more practical suggestions for future research, especially regarding experimental validation of the proposed epitopes.
8. Add more background on the accuracy and reliability of the computational models used for predicting epitopes to provide better context.
9. Focus more on the immunological insights gained from the study rather than just describing the computational methods.
Author Response
Comment:The Manuscript titled “Comprehensive Analysis of the Immune Response to SARS-CoV-2 Epitopes: Unveiling Potential Targets for Vaccine Development” by Deng et al. presents a robust meta-analysis of SARS-CoV-2 epitopes, integrating sequence and structural analyses to identify potential vaccine targets. It is timely and addresses an important topic. However, some concerns must be addressed to increase clarity, scientific consistency, and accessibility.
Response:
Dear Reviewer 1,
Thank you very much for your time to review our manuscript entitled " Comprehensive Analysis of the Immune Response to SARS-CoV-2 Epitopes: Unveiling Potential Targets for Vaccine Development" and your very encouraging comments and suggestions. The comments are very valuable for us to revise and provide important guidance to our research. We agree with your suggestion, and we have made necessary correction according to your recommendations, and we hope it will make our point clearer. Revised portions are marked in yellow color on the track change on manuscript version. Following is point by point response to your comments and concerns.
Comment 1:The manuscript uses sentences that are too complex and includes technical terms like "sequence Shannon entropy" and "structural network correlation," which make it hard to read. "Response frequency (RF) scores" in the method section. Please simplify the language and explain specialized terms early in the text to make the study more straightforward for a wider immunology audience.
Response 1:Thank you for your comment. We have revised and refined the sentences in the manuscript to ensure they are simpler and easier to understand. Additionally, we have provided explanations for technical terms such as "sequence Shannon entropy," "structural network correlation," and "Response Frequency (RF) scores" in the manuscript. The language revisions and edits are highlighted in yellow within the manuscript. The supplementary modifications to some technical terms in the manuscript are as follows:
“sequence Shannon entropy”:Entropy provides a quantification method that can be used to assess the conservation of various positions in a sequence. By calculating the probability of each symbol appearing in a sequence and applying the Shannon entropy formula, the entropy value of the sequence is derived. A low Shannon entropy value indicates that the amino acid or nucleotide at that position is highly conserved across multiple sequences, while a high entropy value indicates a high degree of variation at that position. Compared with sequence conservation analysis based on quantity, sequence conservation analysis based on sequence Shannon entropy considers all possible symbols at a location and their probability distribution, thus reflecting the complexity of variation in the sequence, not just the simple mutation frequency. Compared with analysis based on sequence mutation frequency, it has advantages and is widely used in the analysis of biological sequence conservation and variation.
“structural network degree”: To analyze functionally important residues in the SARS-CoV-2 proteome, we applied a structural network analysis approach. This method utilizes high-quality protein structural data and network theory to identify topologically significant residues. Residues are evaluated based on their direct and indirect local connectivity through non-covalent interactions, their involvement in interaction networks, and their proximity to known protein ligands. This approach has been widely used in studies to identify key functional residues. Residues with high network connectivity are generally considered to exhibit structural centrality, functional importance, and evolutionary constraints (Gaiha et al., 2019).
“Response frequency (RF) scores”: To identify the most researched and frequently recognized epitopes, we used the Immunobrowser tool to calculate the response frequency scores (RFscore) for the epitopes. This index reflects the overall recognition frequency of specific epitopes and their specific residues. For a given epitope, the response frequency is calculated based on the number of individuals tested and the number of individuals tested positive, where R=r/t , with rbeing the total number of responsive donors, and t being the total number of tested donors [14]. Additionally, we calculated the lower and upper bounds of the 95% confidence interval (CI) for the response frequency (RF) at each target protein position. This helps estimate the reliability of the response frequency, determine whether the response frequency is statistically significant, understand and interpret variability and reliability in biological data, and guide further research directions.
Comment 2: Add a brief summary of the current challenges in SARS-CoV-2 vaccine research to highlight how this study contributes to addressing them.
Response 2:We appreciate your comment. We have added a brief summary of the current challenges in SARS-CoV-2 vaccine research to highlight how this study contributes to addressing them in the introduction section:
"Despite advancements in vaccine development and therapeutic strategies, current SARS-CoV-2 vaccines still face numerous challenges, including viral mutability, immune durability, global accessibility, immune evasion, and safety and side effects. There is a need for the development of broadly protective vaccines that can address the current and future COVID-19 pandemics. Achieving these goals necessitates a deeper understanding of the virus's immunological properties to inform future research and vaccine strategies [3–7]. The identification and characterization of antigenic epitopes, which are specific segments of the virus recognized by the immune system, are fundamental to understanding the complexities of host-pathogen interactions and enhancing vaccine effectiveness [8,9]. Such knowledge can further facilitate the development of SARS-CoV-2 vaccines, elucidate SARS-CoV-2-associated immunopathology, and serve as a benchmark in the evaluation of various vaccine candidates. Understanding the antigenic epitope landscape of SARS-CoV-2 and identifying highly conserved and evolutionarily constrained immunodominant antigenic epitopes is of great significance in the development of broad-spectrum vaccines. In this study, we systematically describe the immunological landscape of SARS-CoV-2 based on current research, and through sequence and structural analysis, we screen valuable candidate epitopes to provide valuable guidance for future research."
Comment 3: The manuscript does not discuss the vaccines currently being used for SARS-CoV-2. It would help explain how this study improves existing vaccine strategies and emphasize its importance in tackling emerging variants.
Response 3:We appreciate your comment. We have added the vaccines currently being used for SARS-CoV-2 content in the discussion section: "Existing SARS-CoV-2 vaccines are widely used globally, including mRNA vaccines (such as Pfizer and Moderna), viral vector vaccines (such as AstraZeneca and Johnson & Johnson), and inactivated vaccines (such as Sinovac and Sinopharm). These vaccines have achieved significant success in reducing infection rates and severe disease. However, with the constant mutation of the virus, especially the emergence of new variants that can partially evade immune responses, the effectiveness of existing vaccines may be challenged. Developing broad-spectrum vaccines by targeting highly conserved and evolutionarily constrained immunodominant epitopes may be an effective strategy."
Comment 4: The methods section is detailed but contains many technical terms that can take time to follow. Simplifying or providing extra explanations for terms like "response frequency (RF) scores" would make it easier to understand.
Response 4:We appreciate your comment. As we responded in Comment 1, We have provided simple explanations for technical terms such as "response frequency (RF) scores," "structural network degree," and "sequence Shannon entropy" in the Methods section.
Comment 5: The results cover a lot of information but are presented in an overwhelming way. Breaking them into sections like "Epitope Analysis," "Population Coverage," and "Structural Insights" could improve clarity and readability.
Response 5:We appreciate your comment. Our approach to presenting the results is as follows: First, we provide an overview of the meta-analysis of SARS-CoV-2 antibody and T-cell epitopes. Then, we analyze the distribution characteristics of antibody and T-cell epitopes for each antigen within the SARS-CoV-2 proteome. Based on these distribution characteristics, we conduct a correlation analysis of epitope response frequency, sequence conservation, and structural networking to clarify the conservation and evolutionary traits of antibody and T-cell epitopes. Finally, we identify candidate epitopes that are highly conserved and high-networked degree, and assess their conservation across representative zoonotic coronaviruses and SARS-CoV-2 variants, as well as the population coverage related to T-cell epitopes. These analyses provide valuable insights for further broad-spectrum immunodominant epitope studies. Given the involvement of dozens of antigens in the SARS-CoV-2 proteome, the results may initially seem less readable. We have revised the results section to enhance clarity and readability, with changes highlighted in yellow in the manuscript.
Comment 6: The figures, such as the reactivity maps and network analyses, are complex and challenging to interpret. Consider splitting them into smaller, focused panels and adding clearer explanations in the figure legends.
Response 6:We appreciate your comment. We have reorganized the figures and figure legends in the results section, splitting them into smaller, more focused panels and adding clearer explanations to the figure legends to enhance clarity.
Comment : Include more practical suggestions for future research, especially regarding experimental validation of the proposed epitopes.
Response 7:We appreciate your comment. We have added the following to the discussion section: "Future work should validate the candidate epitopes experimentally to aid vaccine and diagnostic tool development. This involves examining the cross-recognition of these epitopes across variants using immunological assays. Also, their immunogenicity and protection range should be assessed in animal models. Furthermore, based structural biology studies to understand how these epitopes interact with the immune system. And the lastly, ensure no adverse reactions in preclinical toxicology and safety evaluations."
Comment 8: Add more background on the accuracy and reliability of the computational models used for predicting epitopes to provide better context.
Response 8:We appreciate your comment. In this study, the epitope data we utilized are exclusively derived from experimentally validated epitopes, not from computational prediction models. This approach ensures the reliability and accuracy of our data, providing a robust foundation for the development of vaccines and diagnostic tools.
Comment 9: Focus more on the immunological insights gained from the study rather than just describing the computational methods.
Response 9:We appreciate your comment. We have added insights on immunology to the manuscript. The details are as follows:“Our study identifies conserved immunodominant antigenic epitopes across multiple zoonotic coronaviruses and SARS-CoV-2 variants, highlighting their potential as effective immune targets. These findings provide a basis for developing vaccines that elicit broad immune responses. The study also emphasizes the role of these epitopes in interacting with the host immune system, particularly in inducing neutralizing antibodies and T-cell responses. By analyzing epitope conservation across different strains, we offer new insights into immune evasion mechanisms, which are critical for designing vaccines capable of addressing viral mutations. These findings not only guide vaccine development but also serve as a theoretical foundation for future experimental validation.”
We want to take this opportunity to thank you for all your time involved and this great opportunity for us to improve the manuscript. We hope you will find this revised version satisfactory.
Sincerely,
The Authors
Reviewer 2 Report
Comments and Suggestions for Authors
In the manuscript the authors have performed meta-analysis on the epitopes of SARS-CoV-2 to understand its immunological landscape. According to sequence-based correlation analysis, authors found a positive correlation between high viral variability and epitopes that were recognized by B cells and CD4+ T cells which is related to the robust immune response. On the other hand, CD8+ T cells exhibited anti-correlation.
In my opinion, the manuscript is well structured and presented results are relevant to the ongoing studies of SARS-CoV-2. However, I would need to see some revision before accepting this manuscript for publication. My comments are as follows:
1. Elaborate introduction for the better understanding of background of the study.
2.Since the data is collected from multiple sources, it should be discussed how this can cause variability in the results.
3. Separate analysis for different host species is missing. It can lead to better understanding of the effect of the host on the results.
4. There is no discussion on the different studies performed which can include bias in the results.
5. RF score is not reliable all the time, authors should use other scores as well.
6. The study doesn't consider the demographic. It will make more sense to include demographic variation of the results as well.
7.Text in the figures are very hard to read. Increasing the font size is necessary.
Comments on the Quality of English LanguageVery well written. It can still be improved a bit.
Author Response
Comments : In the manuscript the authors have performed meta-analysis on the epitopes of SARS-CoV-2 to understand its immunological landscape. According to sequence-based correlation analysis, authors found a positive correlation between high viral variability and epitopes that were recognized by B cells and CD4+ T cells which is related to the robust immune response. On the other hand, CD8+ T cells exhibited anti-correlation.
In my opinion, the manuscript is well structured and presented results are relevant to the ongoing studies of SARS-CoV-2. However, I would need to see some revision before accepting this manuscript for publication. My comments are as follows:
Response :
Dear Reviewer2,
Thank you very much for your time to review our manuscript entitled " Comprehensive Analysis of the Immune Response to SARS-CoV-2 Epitopes: Unveiling Potential Targets for Vaccine Development" and your very encouraging comments and suggestions. The comments are very valuable for us to revise and provide important guidance to our research. We agree with your suggestion, and we have made necessary modifications according to your recommendations, and we hope it will make our point clearer. Revised portions are marked in yellow color on the track change on manuscript version. Following is point by point response to your comments and concerns.
Comment 1: Elaborate introduction for the better understanding of background of the study.
Response 1:We appreciate your comment. As suggested by Reviewer 1, we have expanded the introduction of our manuscript to include a more detailed background of the study. The specifics are as follows:
- "Despite advancements in vaccine development and therapeutic strategies, current SARS-CoV-2 vaccines still face numerous challenges, including viral mutability, immune durability, global accessibility, immune evasion, and safety and side effects. There is a need for the development of broadly protective vaccines that can address the current and future COVID-19 pandemics. Achieving these goals necessitates a deeper understanding of the virus's immunological properties to inform future research and vaccine strategies [3–7]. The identification and characterization of antigenic epitopes, which are specific segments of the virus recognized by the immune system, are fundamental to understanding the complexities of host-pathogen interactions and enhancing vaccine effectiveness [8,9]. Such knowledge can further facilitate the development of SARS-CoV-2 vaccines, elucidate SARS-CoV-2-associated immunopathology, and serve as a benchmark in the evaluation of various vaccine candidates. Understanding the antigenic epitope landscape of SARS-CoV-2 and identifying highly conserved and evolutionarily constrained immunodominant antigenic epitopes is of great significance in the development of broad-spectrum vaccines. Epitopes of variable pathogens, which are mutationally constrained, are attractive targets for vaccine design, although sequence conservation alone doesn't reliably identify them. Structure-based network analysis has been proven to be a viable strategy. This method applies network theory to protein structure data, quantifying the topological significance of individual amino acid residues,which has been successfully used in numerous studies for the precise identification of evolutionarily constrained sites and antigenic epitopes [11,12]. This approach allowed us to define mutationally constrained antibody and T-cell antigenic epitopes within the SARS-CoV-2 proteome. These epitopes can elicit substantial protective immune responses."
- “ Magazine et al. analyzed SARS-CoV-2 epitopes in the IEDB database, highlighting the impact of Spike protein mutations on immune evasion. They emphasized that predicting mutations and selecting conserved epitopes are crucial for developing broad-spectrum neutralizing therapies or universal vaccines [10].”
- “In this study, we retrieved SARS-CoV-2 epitope datasets from the IEDB and perform a comprehensive meta-analysis, with the aim of illustrating its immunological landscape and identifying highly conserved and evolutionarily constrained immunodominant epitopes. By integrating and analyzing data from various SARS-CoV-2 isolates, we aim to uncover the structural and sequential relationships of these epitopes and their impact on immune recognition. Our study provides an updated overview of immunological data related to COVID-19, highlighting key patterns and identifying specific regions that require further experimental investigation.”
Comment 2: Since the data is collected from multiple sources, it should be discussed how this can cause variability in the results.
Response 2:We appreciate your comment. We have added the following discussion to the discussion section of the manuscript: “The epitope data analyzed from various sources introduces variability due to differences in experimental methods, data quality, and biological factors. The Immune Epitope Database (IEDB) mitigates these challenges through data standardization, quality control, and tools like the ImmunomeBrowser, which visualizes immune responses and knowledge gaps. RFscore prioritizes candidate epitopes based on response frequency, aiding vaccine development, but is limited by data biases, population diversity, and static datasets. To address this, 95% confidence intervals for RF were calculated to assess reliability and guide further research. Future efforts should focus on refining data categorization to extract deeper insights.”
Comment 3: Separate analysis for different host species is missing. It can lead to better understanding of the effect of the host on the results.
Response 3:Thank you very much for your comment. Our epitopes are predominantly derived from Homo sapiens (human) data, accounting for 91.87%. Only a small fraction of epitope data comes from mouse (3.60%) and rhesus macaque (3.35%), with even smaller contributions from other species. The limited host epitope data exhibit substantial variability in RFscore analysis, resulting in low confidence. Consequently, comparative analysis may not yield reliable conclusions. Therefore, we have opted not to analyze these data at this time.
Comment 4: There is no discussion on the different studies performed which can include bias in the results.
Response 4:We appreciate your comment. As responded in Response 2, we have added the discussion on the different studies performed which can include bias in the results in the discussion section.
Comment 5: RF score is not reliable all the time, authors should use other scores as well.
Response 5: We appreciate your comment. In our study, we selected epitopes with cellular activity assay as criteria for epitope screening. We used the epitope response frequency score (RFscore) as a metric to assess how frequently specific epitopes are recognized in experiments. Specifically, it reflects the frequency with which an epitope is recognized under different experimental conditions. By using RFscore, candidate epitopes can be prioritized, aiding researchers in selecting those more likely to play a role in immune responses and those most commonly recognized during natural infection or immunization, thereby enhancing vaccine efficacy. This strategy has been employed in multiple independent studies for characterization. Overall, RFscore is a useful tool for identifying and prioritizing important epitopes in immunological research. Despite its advantages, RFscore has limitations. It relies on existing experimental data and may be influenced by experimental design and conditions. For example, variations in experimental methods and sensitivities across different laboratories can introduce biases. RFscore may not fully reflect genetic diversity in populations, as the data in databases might be predominantly from specific populations or species. Additionally, immune responses are dynamic, and RFscore is a summary based on static data, which may not capture changes in epitope recognition at different stages of infection or disease progression. Lastly, for less-studied epitopes, RFscore might not be accurate due to insufficient data for reliable statistical outcomes. To overcome its limitations, we also calculated the lower and upper bounds of the 95% confidence interval (CI) for the response frequency (RF) at each target protein position (Figure S2). This helps estimate the reliability of the response frequency, determine its statistical significance, and understand and interpret the variability and reliability of biological data, guiding further research directions. To gain a more comprehensive understanding of these data, we also calculated the epitope assay counts, as shown in Figure S1. These results are consistent with the RF analysis findings.
Comment 6: The study doesn't consider the demographic. It will make more sense to include demographic variation of the results as well.
Response 6:We appreciate your comment. In this study, our main goal was to conduct a systematic meta-analysis of the available SARS-CoV-2 immune response epitope data. We aimed to map the immune response landscape of SARS-CoV-2 and identify highly conserved and evolutionarily constrained immunodominant epitopes using sequence conservation and structural network analysis. This work provides valuable insights for developing broad-spectrum vaccines and diagnostic tools in the future. Including demographic variations in the results would add more value. However, we currently lack access to population-specific data. Future studies will focus on further analysis in this area.
Comment 7: Text in the figures are very hard to read. Increasing the font size is necessary.
Response 7: We appreciate your comment. We have modified the font in the figures to make them easier to read.
Comment 8: Comments on the Quality of English Language. Very well written. It can still be improved a bit.
Response 8: We appreciate your comment. We have revised the language of the manuscript, with the modified parts highlighted in yellow within the manuscript.
We want to take this opportunity to thank you for all your time involved and this great opportunity for us to improve the manuscript. We hope you will find this revised version satisfactory.
Sincerely,
The Authors
Reviewer 3 Report
Comments and Suggestions for Authors
This manuscript intends to provide a comprehensive meta-analysis of SARS-CoV-2 antigenic epitopes covering the virus' proteome. The authors integrate sequence variability and residue network connectivity to show their correlations with B- and T-cell response frequencies and try to pinpoint highly conserved and functionally constrained epitopes.
The authors' understanding of basic immunology is limited, and they ignore cellular mechanisms of antigen processing and presentation. Therefore, their data analyses and discussion do not consider the importance of protein localization and processing by cell-surface, endosomal, and cytoplasmic proteases, including proteasomes. Further, they do not consider separate methods of detection and significance levels of T-cell responses.
Major problems and limitations
1. The authors analyzed only linear epitopes. However, when evaluating the reactivity of antiviral antibodies, conformational epitopes, especially those of neutralizing nature, are more important in preventing virus attachment and entry than those reacting with linear epitopes. Variant-resistant or conserved epitopes have been extensively studied through three-dimensional structural analyses (for example, Hastie, KM et al. Science, 2021, 374:472-478; Yan, W. et al. Signal Transduction and Targeted Therapy 2022, 7:26), and those previous results are not consistent with the present authors' predictions. Universal vaccines cannot be designed without knowing the importance of conformational epitopes.
2. T-cell epitopes are linear peptides as they are proteolytically processed from the primary structure of translation products. However, several ways of detecting and quantifying T-cell responses exist, including MHC-peptide tetramer binding, proliferation, expression of activation markers, cytokine production, and ELISPO assays. These responses are quantitative, and one has to set a threshold for a positive response that can differ from one experiment to another. The authors are using response frequencies (RF), but the numbers of individuals showing positive responses to a particular peptide can differ depending on the above detection method and threshold of positivity. This is especially true for epitopes of low response frequency. In this regard, a recent report showing the results of similar analyses using the same IEDB dataset (Magazine, N et al. ImmuneHorizons 2024, 8:214-226) reported only the epitopes showing response frequency exceeding 80% and total number of 10 or more tested individuals. The present authors need to cite this important preceding report. It should be further pointed out that the authors' definition of RF = (r- SQRT(r))/t indicates that the rate can never reach 100%, but in Figure 2, panels A and K, and in Figure 3, panel I, the RF scores often reach 100%. This cannot be true.
3. The authors describe a positive correlation between the epitopes recognized by B and CD4+ T cells. However, this is not specific to SARS-CoV-2 and is expected from antigen processing and presentation mechanisms. Antibodies interact with antigenic structures located on the surfaces of a pathogen or infected cells, and CD4+ T-cells recognize peptides processed from extracellular and cell-surface proteins. Further, for the production of antibodies reactive to proteins, T-B interactions are required, and these interactions depend on the presence of relevant B- and T-cell epitopes on the same antigenic particle. Thus, B- and CD4+ T-cell epitopes often overlap or are located close. On the other hand, epitopes recognized by CD8+ T cells are processed from intracellular proteins and often separate from B- or CD4+ T-cell epitopes. In this regard, the RF score for each residue was calculated by taking a sliding window of 10 residues. Why did the authors take this number? Epitopes recognized by CD8+ T-cells can be shorter, and CD4+ T-cell epitopes are usually longer than ten residues. Linear antigenic epitopes can be shorter. Binding affinities of 8-, 9-, 10- and 11-residue peptides derived from a single oligopeptide can differ drastically.
4. How exactly was the "network score" calculated? Is this a centrality measure? The authors describe that regions with high degrees of variability are associated with elevated RF, but this can be because of the positive selection of escape variants through immune responses. Further, they also describe that CD8+ reactivity positively correlated with the centrality of the residue network. The authors interpret this to be a result of possible favorable recognition of a protein's core areas with crucial functional roles. However, T-cell receptors do not recognize the functionality of a peptide within a protein. Instead, such functionally conserved epitopes might be shared between host and pathogen species and thus become a subject of negative selection in the thymus. An alternative interpretation is that those highly networked residues might be located in regions of a protein preferentially cut into suitable lengths for transportation into ERs and binding to peptide-presenting grooves of MHC class I molecules. More immunologically relevant insights are required here.
The text is filled with grammatic and syntax errors. Just as some examples, what does "the abundant epitopes overlapping structures and homologous sequences from various SARS-CoV-s isolates" mean in the Abstract? The surface glycoprotein and Spike glycoprotein are both used. What does "the average degree-based network value at the highest oligomeric state" indicate? What does "this is bound to be significant redundancy" mean?
Author Response
Comments:This manuscript intends to provide a comprehensive meta-analysis of SARS-CoV-2 antigenic epitopes covering the virus' proteome. The authors integrate sequence variability and residue network connectivity to show their correlations with B- and T-cell response frequencies and try to pinpoint highly conserved and functionally constrained epitopes. The authors' understanding of basic immunology is limited, and they ignore cellular mechanisms of antigen processing and presentation. Therefore, their data analyses and discussion do not consider the importance of protein localization and processing by cell-surface, endosomal, and cytoplasmic proteases, including proteasomes. Further, they do not consider separate methods of detection and significance levels of T-cell responses.
Response : Thank you for the comments and suggestions on the manuscript. We would like to thank you for your professional review work, constructive comments, and valuable suggestions on our manuscript. Your time and efforts are greatly appreciated.
In this study, we aim to provide a comprehensive analysis of SARS-CoV-2 antigenic epitopes across the entire proteome of the virus. We combine sequence variability with structure-based network analysis to understand how these epitopes relate to B cell and T cell responses. Our goal is to identify epitopes that are highly conserved and functionally constrained, offering valuable insights for future research. We focus on epitopes that have been experimentally validated for their function, mapping the immune response landscape. Importantly, the epitopes we selected have been confirmed for their immunogenicity through experiments. For T cell epitopes, they have been shown to be correctly processed and presented by antigens. We chose not to analyze protein processing and localization on the cell surface, endosomes, and cytoplasmic proteases, including the proteasome, for this study. Additionally, variations in T cell response detection methods and significance levels can lead to data variability. As mentioned in our response to Reviewer 2's comment 2, the IEDB data come from peer-reviewed publications indexed in PubMed. The Immune Epitope Database (IEDB) employs several strategies to ensure data reliability, consistency, and usability, thereby reducing the impact of data diversity on research outcomes. These strategies include standardizing data collection and annotation, implementing quality control, integrating and cross-validating data, providing filtering tools, supporting data updates and corrections, distinguishing between predictive tools and experimental data, and ensuring transparency and traceability. By standardizing data processing and enforcing strict quality control, the IEDB effectively addresses the challenges of data diversity. These measures enhance the database's reliability and consistency, offering researchers flexible tools to select and analyze data according to their specific needs. In our study, we selected antigenic epitopes with cellular activity assay as criteria for epitope screening. We used the epitope response frequency score (RFscore) as a metric to assess how frequently specific epitopes are recognized in experiments. Specifically, it reflects the frequency with which an epitope is recognized under different experimental conditions. By using RFscore, candidate epitopes can be prioritized, aiding researchers in selecting those more likely to play a role in immune responses and those most commonly recognized during natural infection or immunization, thereby enhancing vaccine efficacy. This strategy has been employed in multiple independent studies for characterization. Overall, RFscore is a useful tool for identifying and prioritizing important epitopes in immunological research. Despite its advantages, RFscore has limitations. It relies on existing experimental data and may be influenced by experimental design and conditions. For example, variations in experimental methods and sensitivities across different laboratories can introduce biases. RFscore may not fully reflect genetic diversity in populations, as the data in databases might be predominantly from specific populations or species. Additionally, immune responses are dynamic, and RFscore is a summary based on static data, which may not capture changes in epitope recognition at different stages of infection or disease progression. Lastly, for less-studied epitopes, RFscore might not be accurate due to insufficient data for reliable statistical outcomes. To overcome its limitations, we also calculated the lower and upper bounds of the 95% confidence interval (CI) for the response frequency (RF) at each target protein position. This helps estimate the reliability of the response frequency, determine its statistical significance, and understand and interpret the variability and reliability of biological data, guiding further research directions.
Major problems and limitations
Comment 1: The authors analyzed only linear epitopes. However, when evaluating the reactivity of antiviral antibodies, conformational epitopes, especially those of neutralizing nature, are more important in preventing virus attachment and entry than those reacting with linear epitopes. Variant-resistant or conserved epitopes have been extensively studied through three-dimensional structural analyses (for example, Hastie, KM et al. Science, 2021, 374:472-478; Yan, W. et al. Signal Transduction and Targeted Therapy 2022, 7:26), and those previous results are not consistent with the present authors' predictions. Universal vaccines cannot be designed without knowing the importance of conformational epitopes.
Response 1: We appreciate your comment. Conformational epitopes are crucial in designing antibody-based vaccines because neutralizing antibodies often target these epitopes on the virus surface to prevent it from entering host cells. Previous research has extensively studied the conserved or variation-tolerant epitopes of the SARS-CoV-2 spike protein. In our meta-analysis of B cell epitopes, we included both conformational and linear epitopes to understand the immune response better. The results are shown in Table 1 and Table 3 and Supplementary Data Table S3. For our further analysis, we focused on linear B cell epitopes. This is because our vaccine design strategy uses linear epitopes in peptide form, which are easier to develop. These selected linear epitopes have been shown to produce specific antibodies with neutralizing activity. Linear epitopes are also more straightforward to use in some diagnostic tools since they do not depend on the protein's natural shape. Both linear and conformational epitopes are important, depending on the research goals and applications. Conformational epitopes are often vital for generating neutralizing antibodies and vaccine design. In contrast, linear epitopes can be more practical for certain diagnostic tests. The significance of each type varies based on immunological needs and technical applications. Designing vaccines based on conformational epitopes is more complex. In our study, we aimed to identify valuable epitopes and provide new ideas for the rational design of epitope-based vaccines.
Comment 2: T-cell epitopes are linear peptides as they are proteolytically processed from the primary structure of translation products. However, several ways of detecting and quantifying T-cell responses exist, including MHC-peptide tetramer binding, proliferation, expression of activation markers, cytokine production, and ELISPO assays. These responses are quantitative, and one has to set a threshold for a positive response that can differ from one experiment to another. The authors are using response frequencies (RF), but the numbers of individuals showing positive responses to a particular peptide can differ depending on the above detection method and threshold of positivity. This is especially true for epitopes of low response frequency. In this regard, a recent report showing the results of similar analyses using the same IEDB dataset (Magazine, N et al. ImmuneHorizons 2024, 8:214-226) reported only the epitopes showing response frequency exceeding 80% and total number of 10 or more tested individuals. The present authors need to cite this important preceding report. It should be further pointed out that the authors' definition of RF = (r- SQRT(r))/t indicates that the rate can never reach 100%, but in Figure 2, panels A and K, and in Figure 3, panel I, the RF scores often reach 100%. This cannot be true.
Response 2: We appreciate your comment.
- We thoroughly read the article by Magazine, N et al. in ImmuneHorizons 2024, 8:214-226. We agree with your perspective and have cited this paper in our manuscript as an important basis for our analysis. In this study, we aim to provide a comprehensive meta-analysis of SARS-CoV-2 antigenic epitopes, covering the entire viral proteome. We integrate sequence variability with structural network analysis to show the correlation between these epitopes and the frequency of B cell and T cell responses. Our goal is to identify highly conserved and functionally constrained epitopes, offering valuable insights for future research. The epitope based on Meta-analysis primarily aims to elucidate the immune response landscape of the SARS-CoV-2 proteome, providing support for further screening of highly immunogenic candidate antigenic epitopes.
- “It should be further pointed out that the authors' definition of RF = (r- SQRT(r))/t indicates that the rate can never reach 100%, but in Figure 2, panels A and K, and in Figure 3, panel I, the RF scores often reach 100%. This cannot be true.”
Thank you very much for pointing this out. We carefully checked the data, and it was indeed our oversight. We have corrected the results. Currently, there are two methods for calculating RFscore: RF = (r - SQRT(r))/t and RF = r/t. The calculation strategy RF = (r - SQRT(r))/t was applied in the study by Kim et al., 2012. In our research, we analyzed both strategies separately. In the presentation of our research results, we used a simpler calculation strategy, with the formula RF = r/t. Lastly, for less-studied epitopes, RFscore might not be accurate due to insufficient data for reliable statistical outcomes. To overcome its limitations, we also calculated the lower and upper bounds of the 95% confidence interval (CI) for the response frequency (RF) at each target protein position. This helps estimate the reliability of the response frequency, determine its statistical significance, and understand and interpret the variability and reliability of biological data, guiding further research directions.
Comment 3:The authors describe a positive correlation between the epitopes recognized by B and CD4+ T cells. However, this is not specific to SARS-CoV-2 and is expected from antigen processing and presentation mechanisms. Antibodies interact with antigenic structures located on the surfaces of a pathogen or infected cells, and CD4+ T-cells recognize peptides processed from extracellular and cell-surface proteins. Further, for the production of antibodies reactive to proteins, T-B interactions are required, and these interactions depend on the presence of relevant B- and T-cell epitopes on the same antigenic particle. Thus, B- and CD4+ T-cell epitopes often overlap or are located close. On the other hand, epitopes recognized by CD8+ T cells are processed from intracellular proteins and often separate from B- or CD4+ T-cell epitopes. In this regard, the RF score for each residue was calculated by taking a sliding window of 10 residues. Why did the authors take this number? Epitopes recognized by CD8+ T-cells can be shorter, and CD4+ T-cell epitopes are usually longer than ten residues. Linear antigenic epitopes can be shorter. Binding affinities of 8-, 9-, 10- and 11-residue peptides derived from a single oligopeptide can differ drastically.
Response 3: We appreciate your comment.
- Antibodies interact with antigenic structures located on the surfaces of a pathogen or infected cells, and CD4+ T-cells recognize peptides processed from extracellular and cell-surface proteins. Further, for the production of antibodies reactive to proteins, T-B interactions are required, and these interactions depend on the presence of relevant B- and T-cell epitopes on the same antigenic particle. It's important to note that which peptides can effectively be presented to activate CD4+ T cells, and further promote B cell development, depends on the HLA-II allele type. Similarly, the ability to present epitopes that activate CD8+ T cells depend on how well HLA-I alleles bind to specific epitopes. Therefore, whether B cell and CD4+ T cell epitopes overlap or are close, or if CD8+ T cell-recognized epitopes are separate from those of B cells or CD4+ T cells, is still uncertain. We conducted a correlation analysis using the virus's variability index (sequence Shannon entropy) and RFscore. The results showed a positive correlation between B cell and CD4+ T cell epitopes and the sequence variability of the virus, and a negative correlation with the degree of structural network centrality. The underlying mechanisms might be very complex. In this study, we observed this phenomenon based on existing data, which could provide important support for effectively identifying candidate immunodominant antigen epitopes.
- “In this regard, the RF score for each residue was calculated by taking a sliding window of 10 residues. Why did the authors take this number?”
Using a sliding window of 10 residues is not intended for predicting new epitopes. The epitope dataset we used consists of functionally validated epitopes from experiments. The epitope data we analyzed come from multiple sources, which can introduce variability in the results. This variability arises from differences in experimental methods, sample sources, data quality, biological variability, data updates, version differences, and annotation and data handling. Many epitopes may have partial or complete overlap. To identify contiguous dominant regions, RF scores for each residue were recalculated to represent a sliding window of 10 residues. In the IEDB Immunobrowser tool, a sliding window size of 10 residues was chosen because it effectively covers the length of most epitopes, balancing the sensitivity and specificity of predictions while maintaining computational efficiency. This choice is based on biological reasoning and standardized experience, ensuring accuracy and efficiency in epitope identification. This parameter setting has been adopted in several independent studies.
Comment 4: How exactly was the "network score" calculated? Is this a centrality measure? The authors describe that regions with high degrees of variability are associated with elevated RF, but this can be because of the positive selection of escape variants through immune responses. Further, they also describe that CD8+ reactivity positively correlated with the centrality of the residue network. The authors interpret this to be a result of possible favorable recognition of a protein's core areas with crucial functional roles. However, T-cell receptors do not recognize the functionality of a peptide within a protein. Instead, such functionally conserved epitopes might be shared between host and pathogen species and thus become a subject of negative selection in the thymus. An alternative interpretation is that those highly networked residues might be located in regions of a protein preferentially cut into suitable lengths for transportation into ERs and binding to peptide-presenting grooves of MHC class I molecules. More immunologically relevant insights are required here.
Response 4:We appreciate your comment.
- To analyze the functionally important residues in the SARS-CoV-2 proteome, we used a structure-based network analysis approach. This method leverages protein structure data and network theory to identify topologically significant residues. These residues are assessed based on their local connectivity through non-covalent interactions, their role in bridging interactions, and their proximity to known protein ligands. This approach has been employed in multiple studies to identify functionally important residues. Generally, residues with a high degree of network connectivity are considered to have structural centrality, functional importance, and evolutionary constraints (Gaiha et al., 2019) In the study of SARS-CoV-2, identifying these highly networked residues helps pinpoint regions critical to the virus's function that are less likely to mutate. This is particularly important for designing vaccines that can protect against multiple variants.
- Highly variable regions are associated with increased RF scores for B cells and CD4+ T cells, possibly due to positive selection of immune escape variants. This provides valuable evidence for future research on viral immune evasion. We found a positive correlation between CD8+ T cell reactivity and residue network centrality. Residue network centrality refers to highly networked residues located at the core of the protein structure, potentially connecting directly or indirectly with many other residues through non-covalent interactions, forming a complex interaction network. These residues are not necessarily located inside the protein. On the other hand, as you explained, these highly networked residues may be located in regions that are more easily cleaved into suitable lengths for transport to the endoplasmic reticulum and binding to the peptide-binding groove of MHC class I molecules, forming pMHC complexes for antigen presentation and TCR recognition.
- These functionally conserved epitopes were identified by comparing them with human peptide data, revealing that they are not shared between host and pathogen species. As a result, they are not subject to negative selection in the thymus.
Comment 5: The text is filled with grammatic and syntax errors. Just as some examples, what does "the abundant epitopes overlapping structures and homologous sequences from various SARS-CoV-s isolates" mean in the Abstract? The surface glycoprotein and Spike glycoprotein are both used. What does "the average degree-based network value at the highest oligomeric state" indicate? What does "this is bound to be significant redundancy" mean?
Response 5:We appreciate your comment. We have corrected and refined the grammatical and syntactical errors in the manuscript. Additionally, we have addressed the following specific issues:
- "The abundant epitopes overlapping structures and homologous sequences from various SARS-CoV-s isolates"
We found the description unclear and have revised it.
- “The surface glycoprotein and Spike glycoprotein are both used.”
We have standardized the terminology to use "surface glycoprotein" consistently.
- What does "the average degree-based network value at the highest oligomeric state" indicate?
For proteins that function as multimers and in different conformations, we calculated the structural network degree of the residues and normalized these values to assess the extent of structural networking of these residues across different conformational states.
- What does "there is bound to be significant redundancy" mean?
The mean is that the SARS-CoV-2 epitope dataset contains a substantial number of duplicate epitopes or partially overlapping epitopes.
We want to take this opportunity to thank you for all your time involved and this great opportunity for us to improve the manuscript. We hope you will find this revised version satisfactory.
Sincerely,
The Authors
Round 2
Reviewer 1 Report
Comments and Suggestions for Authors
I have reviewed the authors' responses and the changes made to the manuscript. The authors have adequately addressed all the points raised, adding details, improving figures, and discussing limitations as suggested. These changes have improved the clarity of the study. Based on these thorough revisions, I believe the manuscript is ready for publication.
Author Response
Comments:
I have reviewed the authors' responses and the changes made to the manuscript. The authors have adequately addressed all the points raised, adding details, improving figures, and discussing limitations as suggested. These changes have improved the clarity of the study. Based on these thorough revisions, I believe the manuscript is ready for publication.
Response:
Dear Reviewer 1,
Thank you very much for your time to review our manuscript entitled " Comprehensive Analysis of the Immune Response to SARS-CoV-2 Epitopes: Unveiling Potential Targets for Vaccine Development" and your very encouraging comments
Sincerely,
The Authors
Reviewer 3 Report
Comments and Suggestions for Authors
The manuscript has been modified partly following this reviewer's comments. However, some scientifically important points must still be addressed, and the new sentences added during the revision contain immunologically misleading descriptions, which must be rectified.
1. The authors' usage of SARS-CoV-2 protein names is inconsistent with standard nomenclature. The S (glyco)protein of coronaviruses is called the Spike (Alexander SP et al. Coronavirus (CoV) proteins in GtoPdb v.2024.2. IUPHAR/BPS Guide to Pharmacology CITE. 2024: https://doi.org/10.2218/gtopdb/F1034/2024.2), not a "Surface" glycoprotein as the authors repeatedly describe. The authors' descriptions are even more confusing as they sometimes use "surface glycoprotein" instead of "Surface." In SARS-CoV-2, the M, E, and N proteins are also glycosylated in addition to the S (Yanqiu G et al. The glycosylation in SARS-CoV-2 and its receptor ACE2. Signal Transduction and Targeted Therapy. 2021, 6:396; Aloor A et al. Glycosylation in SARS-CoV-2 variants: A path to infection and recovery. Biochem Pharmacol. 2022, 206:115335), and M and E proteins are located on the surface of the virions. Thus, SARS-CoV-2 "surface glycoprotein" can also mean M or E "glycoproteins."
Further, ORF1a and ORF1b overlap each other in the SARS-CoV-2 genome and the polyprotein pp1ab is translated by ribosomal frameshifting at the C-terminus of pp1a. Thus, the authors' sliding windows of 10 residues encompass this frame-shifting site, and about 20 such 10-mer peptides are not canonically translated in infected cells. Given this frameshifting, how do the authors relate the reduced T-cell response rates to the RdRp (Nsp12)? In addition, due to the use of the pp1ab sequence but not that of pp1a, the C-terminus of pp1a, the Nsp11, is not included in this "comprehensive" analysis.
2. In the yellow-highlighted, added paragraphs, the authors repeatedly describe "preventing viral infections" or "protective immune responses" in relation to B- and T-cell epitopes. However, not all T-cell responses are protective. In fact, in SARS-CoV-2 infection, some T helper cell responses are associated with tissue injury and the pathogenesis of pneumonia. Further, CD8+ CTL responses are inevitably associated with eliminating virus-infected cells, which can destroy the epithelial lining and sometimes endothelial cells, causing fatal pathology. As CD8+ T cells recognize peptides produced inside infected cells, their activation occurs only after infection. Thus, CTL responses can be "protective" only when they develop rapidly enough while the number of infected cells is still small, so that their destruction does not cause pathogenesis.
Antibody responses can be protective by preventing the spread of the virus within an infected individual. However, true prevention can only be achieved by inducing neutralizing antibodies on the surfaces of the virus entry site. This sterilizing immunity can only be achieved by the long-term production of secretory IgA, which cannot be accomplished by merely identifying B-cell epitopes. Thus, the authors' understanding and descriptions of "immune protection" are too naive and often result in overstatements concerning the present analyses.
3. The text is still filled with repetitions and syntax errors. In section 3.2 on page 7, "the entire protein of surface glycoprotein" is redundant, not to mention the above inappropriateness of the "surface glycoprotein." On page 9, "the reactivity of antibody and T cell responses" appears 4 times in a single paragraph. Further, "reactivity of responses" does not make sense. What mechanisms do the authors indicate by "the different mechanisms induced by antibody, CD4+, and CD8+ T cell-mediated immune responses?" On page 11, what does "a balance among these critical parameters must be struck" mean? On page 15, the "CTL T cell" is redundant. In section 3.5 on the same page, what do the authors indicate by "our mouse-based animal model?" Is this published?
In the Discussion, existing SARS-CoV-2 vaccines include recombinant proteins like the Novavax one. On page 17, how can the authors be sure that their study is associated with the induction of neutralizing antibodies? They studied only B-cell epitopes, which are not necessarily associated with virus-neutralizing potentials.
Author Response
Comments:
The manuscript has been modified partly following this reviewer’s comments. However, some scientifically important points must still be addressed, and the new sentences added during the revision contain immunologically misleading descriptions, which must be rectified.
Response:
Dear Reviewer 3,
Thank you for the comments and suggestions on the manuscript. We would like to thank you for your professional review work, constructive comments, and valuable suggestions on our manuscript. Your time and efforts are greatly appreciated. The comments are very valuable for us to revise and provide important guidance to our research. We agree with your suggestion, and we have made necessary modifications according to your recommendations, and we hope it will make our point clearer. Revised portions are marked in green color on the track change on manuscript version. Following is point by point response to your comments and concerns.
Comments 1:
- The authors' usage of SARS-CoV-2 protein names is inconsistent with standard nomenclature. The S (glyco)protein of coronaviruses is called the Spike (Alexander SP et al. Coronavirus (CoV) proteins in GtoPdb v.2024.2. IUPHAR/BPS Guide to Pharmacology CITE. 2024: https://doi.org/10.2218/gtopdb/F1034/2024.2), not a "Surface" glycoprotein as the authors repeatedly describe. The authors' descriptions are even more confusing as they sometimes use "surface glycoprotein" instead of "Surface." In SARS-CoV-2, the M, E, and N proteins are also glycosylated in addition to the S (Yanqiu G et al. The glycosylation in SARS-CoV-2 and its receptor ACE2. Signal Transduction and Targeted Therapy. 2021, 6:396; Aloor A et al. Glycosylation in SARS-CoV-2 variants: A path to infection and recovery. Biochem Pharmacol. 2022, 206:115335), and M and E proteins are located on the surface of the virions. Thus, SARS-CoV-2 "surface glycoprotein" can also mean M or E "glycoproteins." Further, ORF1a and ORF1b overlap each other in the SARS-CoV-2 genome and the polyprotein pp1ab is translated by ribosomal frameshifting at the C-terminus of pp1a. Thus, the authors' sliding windows of 10 residues encompass this frame-shifting site, and about 20 such 10-mer peptides are not canonically translated in infected cells. Given this frameshifting, how do the authors relate the reduced T-cell response rates to the RdRp (Nsp12)? In addition, due to the use of the pp1ab sequence but not that of pp1a, the C-terminus of pp1a, the Nsp11, is not included in this "comprehensive" analysis.
Response 1:
- We are immensely grateful for your professional review of our article. We have revised "surface glycoprotein" to "spike glycoprotein" in the manuscript, as highlighted in green.
- Thank you very much for your comments. For each non-structural protein (nsp1-16 proteins) encoded by ORF1a and ORF1b in the pp1ab polyprotein, we individually mapped the antigenic epitopes of each non-structural protein. These identified antigenic epitopes were mapped to their respective proteins, and the results were displayed in Figure A1 in Appendix A. For a more intuitive presentation, we summarized these individual proteins in the form of the pp1ab polyprotein and presented them as a complete schematic (Figure 2A). The reason nsp11 protein is not shown is because there are currently no research records on the epitopes of the nsp11 protein, so we did not include this protein in the results.
Comments 2:
- In the yellow-highlighted, added paragraphs, the authors repeatedly describe "preventing viral infections" or "protective immune responses" in relation to B- and T-cell epitopes. However, not all T-cell responses are protective. In fact, in SARS-CoV-2 infection, some T helper cell responses are associated with tissue injury and the pathogenesis of pneumonia. Further, CD8+ CTL responses are inevitably associated with eliminating virus-infected cells, which can destroy the epithelial lining and sometimes endothelial cells, causing fatal pathology. As CD8+ T cells recognize peptides produced inside infected cells, their activation occurs only after infection. Thus, CTL responses can be "protective" only when they develop rapidly enough while the number of infected cells is still small, so that their destruction does not cause pathogenesis.
Antibody responses can be protective by preventing the spread of the virus within an infected individual. However, true prevention can only be achieved by inducing neutralizing antibodies on the surfaces of the virus entry site. This sterilizing immunity can only be achieved by the long-term production of secretory IgA, which cannot be accomplished by merely identifying B-cell epitopes. Thus, the authors' understanding and descriptions of "immune protection" are too naive and often result in overstatements concerning the present analyses.
Response 2:
Thank you very much for your professional review of our article. Antigenic epitopes are specific parts of an antigen molecule that are recognized and bound by the immune system, playing a key role in immune responses. They activate immune cells by binding to the receptors of B cells and T cells, inducing antibody production or cytotoxic reactions, and play an important role in the formation of immune memory. This mechanism is the basis for the specificity and effectiveness of the immune system and is crucial for vaccine design, enabling the body to effectively recognize and clear pathogens. However, whether the host's immune system can play a protective role against viruses is a very complex process involving many factors. The immune response is controlled by a variety of factors in terms of quality and quantity, including the form and entry route of the antigen, the nature of professional antigen-presenting cells, the nature of immune cells, the nature of cytokines, the individual's genetic background, and contact history with the antigen. Antigenic epitopes are an important part of the immune recognition characteristics in immune responses. Our findings provide important insights for future research and clinical applications. Our study provides an updated overview of immunological data related to COVID-19, highlighting key patterns and identifying specific regions that require further experimental investigation. Indeed, as you mentioned, it is inappropriate to understand candidate epitopes directly as "immune protection". We have revised sentences in the manuscript that describe "preventing viral infections" or "protective immune responses" to ensure that we do not overinterpret the result data. The revision is as follows: "These epitopes have the potential to elicit substantial protective immune responses."
Comments 3:
- The text is still filled with repetitions and syntax errors. In section 3.2 on page 7, "the entire protein of surface glycoprotein" is redundant, not to mention the above inappropriateness of the "surface glycoprotein."
On page 9, "the reactivity of antibody and T cell responses" appears 4 times in a single paragraph. Further, "reactivity of responses" does not make sense. What mechanisms do the authors indicate by "the different mechanisms induced by antibody, CD4+, and CD8+ T cell-mediated immune responses?"
On page 11, what does "a balance among these critical parameters must be struck" mean?
On page 15, the "CTL T cell" is redundant.
In section 3.5 on the same page, what do the authors indicate by "our mouse-based animal model?" Is this published?
In the Discussion, existing SARS-CoV-2 vaccines include recombinant proteins like the Novavax one.
On page 17, how can the authors be sure that their study is associated with the induction of neutralizing antibodies? They studied only B-cell epitopes, which are not necessarily associated with virus-neutralizing potentials.
Response 3:
- We greatly appreciate your professional review of our article. According to your suggestions, we have made a revision to the sentence, changing "the entire protein of surface glycoprotein" to "The spike glycoprotein".
- We have revised the ambiguous description of "the reactivity of antibody and T cell responses" and changed it to "the antibody and T cell responses".
- What mechanisms do the authors indicate by "the different mechanisms induced by antibody, CD4+, and CD8+ T cell-mediated immune responses?"
This refers to the mechanism where the activation of B cell immune responses requires the assistance of CD4+ T cells, and these two have a functional dependency in immune responses. However, the functional activation of CD8+ T cells does not strongly depend on B cells and CD4+ T cells. This sentence could potentially cause confusion in the interpretation of our results in the manuscript, so we have decided to remove it.
- “On page 11, what does "a balance among these critical parameters must be struck" mean?
This is mean that according to our analysis results, it seems impossible to screen for both highly conserved and evolutionarily constrained B cell and CD4+ T cell epitopes in SARS-CoV-2 variants. It is necessary to find a balance between these two, ensuring that these B cell and CD4+ T cell epitopes make trade-offs among some key genotypes of strains, and obtain as much conserved coverage as possible in key variant strains. We have revised this sentence to make it more concise and easier to understand, as follows: "To screen for highly conserved and evolutionarily constrained B cell and CD4+ T cell epitopes, a balance among these critical parameters must be struck."
- Thank you very much for your comment. We have removed the term "CTL T cell".
- Thank you very much for your comment. We have initially validated these epitopes in mouse experiments, but the result data has not yet been published. To avoid unnecessary misunderstanding, we have removed the word "our".
- We sincerely appreciate the valuable comments. We have supplemented information about the SARS-CoV-2 recombinant protein vaccine and added references into the Discussion part in the revised manuscript.
- The candidate linear B-cell antigen epitopes we screened have been proven to have the ability to induce neutralizing antibodies in existing research, the results can be seen in Supplementary Data Table S4.
We want to take this opportunity to thank you for all your time involved and this great opportunity for us to improve the manuscript. We hope you will find this revised version satisfactory.
Sincerely,
The Authors